# FMIP: Joint Continuous-Integer Flow for Mixed-Integer Linear Programming

**Hongpei Li[1], Hui Yuan[2], Han Zhang[3], Jianghao Lin[4*], Dongdong Ge[4,5], Mengdi Wang[2], Yinyu Ye[4,5,6]**

[1]Shanghai University of Finance and Economics, Shanghai, China
[2]Princeton University, New Jersey, USA
[3]National University of Singapore, Singapore
[4]Antai College of Economics and Management, Shanghai Jiao Tong University, Shanghai, China
[5]Shanghai Institute for Mathematics and Interdisciplinary Sciences, Shanghai, China
[6]Stanford University, California, USA
`ishongpeili@gmail.com, linjianghao@sjtu.edu.cn`

## Abstract

Mixed-Integer Linear Programming (MILP) is a foundational tool for complex decision-making problems. However, the NP-hard nature of MILP presents a significant computational challenge, motivating the development of machine learning-based heuristic solutions to accelerate downstream solvers. While recent generative models have shown promise in learning powerful heuristics, they suffer from a critical limitation. That is, they model the distribution of *only the integer variables* and fail to capture the intricate coupling between integer and continuous variables, creating an information bottleneck and ultimately leading to suboptimal solutions. To this end, we propose Joint Continuous-Integer Flow for Mixed-Integer Linear Programming (**FMIP**), which is the first generative framework that models the *joint distribution* of both integer and continuous variables for MILP solutions. Built upon the joint modeling paradigm, a holistic guidance mechanism is designed to steer the generative trajectory, actively refining solutions toward optimality and feasibility during the inference process. Extensive experiments on eight standard MILP benchmarks demonstrate the superior performance of FMIP against existing baselines, reducing the primal gap by 41.34% on average. Moreover, we show that FMIP is fully compatible with arbitrary backbone networks and various downstream solvers, making it well-suited for a broad range of real-world MILP applications. Our code is available[*].

## 1 Introduction

Mixed-Integer Linear Programming (MILP) represents a cornerstone of mathematical optimization, providing a powerful framework for modeling complex decision-making problems that involve both discrete choices and continuous quantities Zhou et al. (2025); Kratica et al. (2014); He et al. (2015). Its ability to capture the intricate interplay between integer and continuous variables makes it indispensable across diverse domains, including combinatorial optimization (Della Croce and Paschos, 2014), energy systems (Miehling et al., 2023), and supply chain design (Ivanov et al., 2022).

Despite its versatility, solving a MILP instance remains a fundamental challenge due to its NP-hard property (Bertsimas and Tsitsiklis, 1997). Consequently, exact solvers often rely on heuristics to find high-quality solutions in a practical amount of time. This has motivated a surge of interest in using deep learning to predict powerful heuristics that can warm-start and guide the downstream solvers (Nair et al., 2020; Han et al., 2023; Liu et al., 2025; Chen et al., 2025). Among them, generative modeling has emerged as a particularly promising approach. Modern techniques like diffusion models and flow matching reframe the difficult one-step task of predicting a complete solution into an iterative refinement process (Lin et al., 2024b; Sun and Yang, 2023; Feng et al., 2024; Zeng et al., 2024). By learning to transform a simple noise distribution into a distribution of

---

[*]Jianghao Lin is the corresponding author.
[*]https://github.com/Lhongpei/FMIP

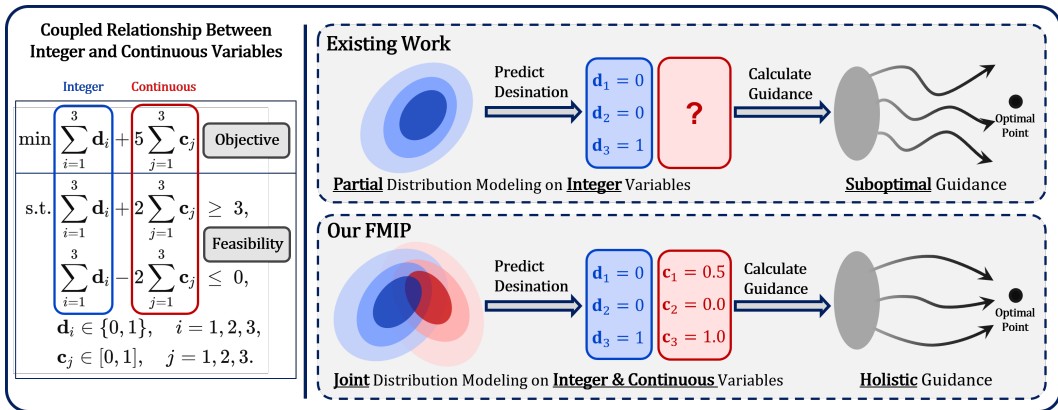

Figure 1: The key advantages of our FMIP over existing works: 1) the joint distribution modeling on both integer and continuous variables and 2) the consequent holistic guidance during inference.

high-quality solutions, these models can effectively navigate the complex combinatorial search space, breaking down the challenge of satisfying intricate constraints into a series of more manageable steps.

However, existing generative methods for MILP suffer from a critical limitation: they model the distribution of **only the integer variables**. As illustrated in Figure 1, this architectural choice ignores the strong, coupled relationship between integer and continuous variables, which is fundamental to the structure of MILP problems. Since both variable types are often required to evaluate the objective function and check for constraint satisfaction, this incomplete modeling creates an information bottleneck, leading to suboptimal solution quality and hindering the effectiveness of any guidance mechanism.

To this end, we propose Joint Continuous-Integer Flow for Mixed-Integer Linear Programming (**FMIP**), which is the first generative framework that models the **joint distribution** of both integer and continuous variables for MILP solutions. FMIP leverages a conditional flow matching process to progressively generate a complete solution, fully capturing the interdependence between all decision variables. This joint modeling approach unlocks a **holistic guidance** mechanism that can steer the generation process using complete, instance-wise feedback from both the objective function and constraint violations. We highlight that while the transition from ILP to MILP appears intuitive, joint modeling represents a critical but previously overlooked gap. Bridging this gap is crucial: 1) it enables us to shift from single-point prediction to generative distribution modeling, addressing the inherent "one-to-many" nature of MILP solutions (multiple optima), and 2) it enables the utilization of instance-wise optimization information, addressing a critical limitation of partial modeling schemes. FMIP is **fully compatible** with arbitrary backbone networks and downstream solvers, showing 41.34% relative improvement on average across eight standard MILP benchmarks over SOTA baselines. Our contributions are summarized as follows:

- We introduce FMIP, the first generative framework to jointly model the complete distribution of both integer and continuous variables for MILP solutions, effectively capturing their critical coupling relationship.
- We design a holistic guidance mechanism that leverages the jointly generated variables to steer the sampling process toward solutions with better objective values and constraint satisfaction.
- As a powerful generative learning paradigm, FMIP is agnostic to the choice of backbone networks and downstream solvers, making it well-suited for a broad range of real-world MILP applications.
- FMIP sets a new state-of-the-art for learning-based MILP heuristics, achieving superior results on eight benchmarks while showing compatibility with diverse backbones and solvers.

## 2 RELATED WORKS

Machine learning is widely used to accelerate Mixed-Integer Linear Programming (MILP) solvers. One major line of research focuses on enhancing the internal components of the Branch-and-Bound algorithm, such as learning policies for variable branching (Gasse et al., 2019; Khalil et al., 2016; Gupta et al., 2020; Zarpellon et al., 2021; Gupta et al., 2022; Scavuzzo et al., 2022; Lin et al., 2024a;

Zhang et al., 2024) or cutting plane selection (Tang et al., 2020; Huang et al., 2022). Another closely related direction, which our work belongs to, aims to predict high-quality heuristic solutions to warm-start the optimization process, thereby reducing the search space and speeding up convergence (Nair et al., 2020; Han et al., 2023; Huang et al., 2024b; Liu et al., 2025; Chen et al., 2025; Zeng et al., 2024).

Most existing solution-prediction methods are *discriminative* in nature, employing models like Graph Neural Networks (GNNs) to predict the variable assignments in a single forward pass (Huang et al., 2024b; Liu et al., 2025; Hu et al., 2024; Huang et al., 2024a; Ye et al., 2024). However, predicting a complete, feasible, and high-quality solution in one step is exceptionally challenging due to the complex combinatorial structure of MILPs. This difficulty has motivated the use of *generative models*, such as diffusion models and flow matching, for MILP heuristics (Feng et al., 2024; Zeng et al., 2024). These models reframe the solution prediction as an iterative refinement process, decomposing the hard single-step task into a more manageable multi-step generation, which is better suited for navigating complex constrained spaces (Zeng et al., 2024). However, we identify a fundamental limitation in these state-of-the-art approaches: prior approaches generally focus on partial distribution modeling on integer variables, which inherently precludes capturing the critical interdependence between all decision variables, leading to suboptimal guidance and inferior solutions. In contrast, our proposed Joint Continuous-Integer Flow for MILP is the first generative framework that explicitly models the joint distribution of both continuous and integer variables, directly addressing this critical gap in the literature.

## 3 PRELIMINARIES

### 3.1 MILP DEFINITION AND GRAPH REPRESENTATION

Suppose we have $n$ decision variables denoted as $\boldsymbol{x} = (\boldsymbol{x}_{\mathcal{I}}, \boldsymbol{x}_{\mathcal{C}}) \in \mathbb{Z}^q \times \mathbb{R}^{n-q}$, where $\boldsymbol{x}_{\mathcal{I}}$ and $\boldsymbol{x}_{\mathcal{C}}$ represent the integer and continuous variables, respectively. An MILP instance can be defined as:

$$
\begin{aligned}
\min_{\boldsymbol{x}} \quad & \boldsymbol{w}^\top \boldsymbol{x} \\
\text{s.t.} \quad & \boldsymbol{A}\boldsymbol{x} \leq \boldsymbol{b} \\
& \boldsymbol{l} \leq \boldsymbol{x} \leq \boldsymbol{u} \\
& \boldsymbol{x}_{\mathcal{I}} \in \{0, 1, \ldots, K\}^q \\
& \boldsymbol{x}_{\mathcal{C}} \in \mathbb{R}^{n-q}
\end{aligned}
\tag{1}
$$

Here, $\boldsymbol{w}$ is the objective function coefficient vector. $\boldsymbol{l}$ and $\boldsymbol{u}$ are the lower bounds and upper bounds for decision variables. $\boldsymbol{A} \in \mathbb{R}^{m \times n}$ and $\boldsymbol{b} \in \mathbb{R}^m$ denote the coefficient matrix and the right-hand-side vector of linear constraints. Moreover, $K$ is a scalar integer defining the maximum feasible value of integer variables (i.e., $\boldsymbol{x}_i \in \{0, 1, \ldots, K\}, \forall i \in \mathcal{I}$). An MILP instance with $m$ linear constraints and $n$ variables can thus be represented by the tuple $\boldsymbol{M} = (\boldsymbol{A}, \boldsymbol{b}, \boldsymbol{l}, \boldsymbol{u}, \boldsymbol{w})$. A solution $\boldsymbol{x}$ is considered feasible if all constraints are satisfied. Note that, in this paper, we focus on MILP problems with *bounded integer variables*, as most real-world applications exhibit this characteristic. Specially, when $K = 1$ such that $\boldsymbol{x}_{\mathcal{I}} \in \{0, 1\}^q$, the problem reduces to Mixed Integer Binary Programming (MIBP), which is the most commonly studied subclass of MILP (Huang et al., 2024b; Liu et al., 2025). Our FMIP framework can naturally extend to unbounded integer variables by incorporating the bit-wise prediction strategy employed in prior studies (Nair et al., 2020; Han et al., 2023). Alternatively, we can continuously relax a subset of these variables to predict their continuous embeddings, which are subsequently discretized via rounding or projection to obtain valid integer assignments.

**Graph Representation for MILP**. When applying deep learning to MILP, it is common to represent an MILP instance as a **bipartite graph** (Gasse et al., 2019). The bipartite graph consists of two sets of nodes: one representing the $n$ decision variables and the other representing the $m$ constraints. Edges in this graph connect variables to the constraints in which they appear, effectively encoding the sparsity pattern of the constraint matrix. Building on this foundation, as shown in the top-left part of Figure 2, we maintain the fundamental bipartite topology but explicitly distinguish between integer and continuous variables as distinct node types. Specifically, an MILP instance is denoted as a graph $\boldsymbol{G} = (\mathcal{V}_{\mathcal{I}}, \mathcal{V}_{\mathcal{C}}, \mathcal{V}_{\text{con}}, \mathcal{E})$, where the nodes are partitioned into three distinct sets:

- **Integer Variable Nodes** ($\mathcal{V}_{\mathcal{I}}$): One node for each of the $q$ integer variables.
- **Continuous Variable Nodes** ($\mathcal{V}_{\mathcal{C}}$): One node for each of the $n - q$ continuous variables.

- **Constraint Nodes** ($\mathcal{V}_{\text{con}}$): One node for each of the $m$ linear constraints.

An edge is drawn between a variable node $j \in \mathcal{V}_{\mathcal{I}} \cup \mathcal{V}_{\mathcal{C}}$ and a constraint node $i \in \mathcal{V}_{con}$ if the variable $x_j$ has a non-zero coefficient in the $i$-th constraint. Based on the rich, relational MLIP graph, various graph neural networks (e.g., GCN (Kipf, 2016) or GAT (Veličković et al., 2017)) can be employed to extract the rich topological representations for later discriminative or generative learning processes. More details can be found in Appendix B and Appendix C.

## 3.2 FLOW MATCHING

Flow Matching (FM) is a powerful framework for training generative models (Lipman et al., 2023; Albergo and Vanden-Eijnden, 2023; Liu et al., 2023). The core idea is to learn a time-dependent vector field $v_t$ that transforms samples from a simple prior distribution $p_0$ (e.g., Gaussian noise) into samples from a target data distribution $p_1$. This transformation is defined by a probability path $(p_t)_{t \in [0,1]}$ interpolating between $p_0$ and $p_1$, which is induced by an Ordinary Differential Equation (ODE): $\frac{d\boldsymbol{c}_t}{dt} = \boldsymbol{v}_t(\boldsymbol{c}_t)$. FM allows the vector field $\boldsymbol{v}_t$ to be learned directly via a simple regression objective, given a defined path between noise and data samples. FM is flexible and can model both continuous and discrete data, which is essential for our work.

- For continuous data, the prior $p_0$ is typically a standard Gaussian $\mathcal{N}(\mathbf{0}, \mathbf{I})$. The interpolating path between data point $\boldsymbol{c}_1$ and noise $\boldsymbol{c}_0$ is defined as $\boldsymbol{c}_t | \boldsymbol{c}_1 \sim \mathcal{N}(t\boldsymbol{c}_1, (1-t)^2 \mathbf{I})$. This yields a simple, closed-form target vector field for model training, i.e., $\boldsymbol{v}_t(\boldsymbol{c}_t | \boldsymbol{c}_1) = \frac{\boldsymbol{c}_1 - \boldsymbol{c}_t}{1-t}$.

- For discrete data, the prior $p_0$ is a uniform categorical distribution over $K$ categories. The conditional path for each component $i$ of a data point $\boldsymbol{d}_1$ is defined as $\boldsymbol{d}_t^{(i)} | \boldsymbol{d}_1^{(i)} \sim \mathrm{Cat}(t\delta(\boldsymbol{d}_t^{(i)}, \boldsymbol{d}_1^{(i)}) + (1-t)/K)$, where $\delta$ is the Kronecker delta and $\mathrm{Cat}(\cdot)$ denotes the categorical distribution. This path implies a target conditional rate matrix $R_{t|1}(\cdot, \cdot | \boldsymbol{d}_1^{(i)})$ for model training:

$$R_{t|1}(\boldsymbol{d}_t^{(i)}, j \mid \boldsymbol{d}_1^{(i)}) = \frac{\delta(\boldsymbol{d}_1^{(i)}, j)}{1-t} \left(1 - \delta(\boldsymbol{d}_1^{(i)}, \boldsymbol{d}_t^{(i)})\right). \tag{2}$$

During inference, these targets are replaced by the model's predictions to generate new samples by evolving the state over time. At each step, the learned model provides an estimate of the vector field $\hat{\boldsymbol{v}}$ or rate matrix $\hat{R}$, which is used to update the current state:

$$\begin{aligned}
\boldsymbol{c}_{t+\Delta t} &= \boldsymbol{c}_t + \hat{\boldsymbol{v}}_{t|1}(\boldsymbol{c}_t | \boldsymbol{c}_1)\Delta t, &&\text{for continuous data,} \\
\boldsymbol{d}_{t+\Delta t}^{(i)} &\sim \mathrm{Cat}\left(\delta(\boldsymbol{d}_{t+\Delta t}^{(i)}, \boldsymbol{d}_t^{(i)}) + \hat{R}_{t|1}(\boldsymbol{d}_t^{(i)}, \boldsymbol{d}_{t+\Delta t}^{(i)})\Delta t\right), &&\text{for discrete data.}
\end{aligned} \tag{3}$$

## 4 FMIP: JOINT CONTINUOUS-INTEGER FLOW FOR MILP

In this section, we introduce FMIP, a generative framework that learns the joint distribution of integer and continuous variables to find high-quality solutions for MILP instances. This unified modeling is the cornerstone of our approach, explicitly designed to capture variable coupling and enable the utilization of instance-wise optimization information, which is inaccessible to partial modeling. As shown in Figure 2, FMIP is built on a time-dependent graph representation, a joint continuous-integer flow model, a holistic guidance mechanism for inference, and seamless integration with downstream solvers. Notably, our FMIP framework is independent of the choice of generative model. We employ flow matching in this work due to its conceptual simplicity and training stability.

### 4.1 TIME-DEPENDENT GRAPH REPRESENTATION

FMIP operates on a time-dependent solution graph, which captures both the static structure of an MILP instance and its dynamic state during the generative process. As defined in Section 3.1, the static structure is encoded in a graph $\boldsymbol{G} = (\mathcal{V}_{\mathcal{I}}, \mathcal{V}_{\mathcal{C}}, \mathcal{V}_{\text{con}}, \mathcal{E})$.

During the flow matching process, we define the solution graph at time-step $t \in [0, 1]$ as the tuple $\boldsymbol{G}_t = (\boldsymbol{G}, \boldsymbol{d}_t, \boldsymbol{c}_t)$, where $\boldsymbol{d}_t$ and $\boldsymbol{c}_t$ are the values of the integer and continuous variables, respectively. This dynamic state information is incorporated directly into the graph by augmenting the static features of each variable node. We define $\boldsymbol{x}_t = (\boldsymbol{d}_t, \boldsymbol{c}_t)$ as the vector of solution values for all variables. $\boldsymbol{x}_t^{(i)}$ denotes the state of the $i$-th variable at time step $t$.

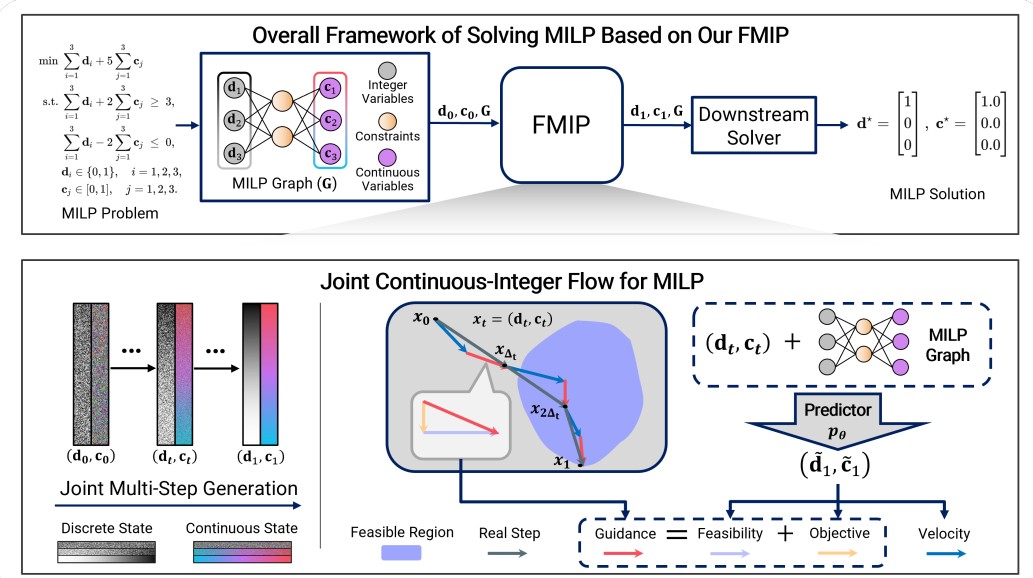

Figure 2: The overall framework of our proposed FMIP.

## 4.2 JOINT CONTINUOUS-INTEGER FLOW

To overcome the limitations of integer-only models, we design a flow matching process over the mixed solution space of both continuous and integer variables. The process constructs a probability path $p_{t|1}(\boldsymbol{G}_t|\boldsymbol{G}_1)$ for $t \in [0, 1]$, which transforms a simple noise distribution $p_0(\boldsymbol{G}_0)$ into the target distribution of high-quality solutions $p_1(\boldsymbol{G}_1)$.

**Joint Training Objective** Our model, parameterized by $\theta$, learns this transformation by predicting the target solution $(\boldsymbol{d}_1, \boldsymbol{c}_1)$ from a noisy state $\boldsymbol{G}_t$. It features two prediction heads over a shared learnable graph backbone network (e.g., GCN or GAT). One head outputs the denoised continuous values $\hat{\boldsymbol{c}}_1(\boldsymbol{G}_t)$ and another outputs the probability distribution of the integer solution $\hat{p}(\boldsymbol{d}_1|\boldsymbol{G}_t)$. The model is trained by minimizing a joint loss function that combines a regression loss for the continuous variables and a cross-entropy loss for the integer variables:

$$\mathcal{L}(\theta) = \mathbb{E}_{t, \boldsymbol{G}_1}\left[\frac{\|\hat{\boldsymbol{c}}_1(\boldsymbol{G}_t) - \boldsymbol{c}_1\|_2^2}{1-t} - \omega \log \hat{p}(\boldsymbol{d}_1|\boldsymbol{G}_t)\right], \tag{4}$$

where $t \sim \mathcal{U}(0, 1)$, $\boldsymbol{G}_1 \sim p_{\text{data}}$, and $\omega$ is a hyperparameter balancing the two terms.

**Sampling Process** At inference, we generate a solution by simulating the forward ODE from $t = 0$ to $t = 1$. At each time step $t$, the model takes the current noisy graph $\boldsymbol{G}_t$ as input and produces its predictions, i.e., $\hat{\boldsymbol{c}}_1(\boldsymbol{G}_t)$ and $\hat{p}(\boldsymbol{d}_1|\boldsymbol{G}_t)$. As illustrated in Eq. 3, these outputs are used to estimate the conditional vector field $\boldsymbol{v}_t$ and rate matrix $R_{t|1}$, enabling the next step of the simulation. We use a cosine schedule for time discretization, which allocates more steps to the low-noise region near $t = 1$ for better quality (Nichol and Dhariwal, 2021). This generative formulation is a key advantage over traditional one-step discriminative predictors, as it enables the multi-step iterative refinement with a holistic guidance mechanism during sampling, which we detail next.

## 4.3 HOLISTIC GUIDANCE MECHANISM

A key benefit of FMIP is that the model predicts a *complete* solution candidate $(\hat{\boldsymbol{d}}, \hat{\boldsymbol{c}})$ at any step $t$ during generation. Hence, holistic guidance can be directly derived using instance-specific information from the MILP formulation itself (both the objective function and constraints), which steers the generation process towards solutions that are both feasible and optimal. Such a capability is absent in previous works that only consider integer variables.

**Guidance Target Function**. We define a target function $f(\boldsymbol{x})$ that captures the two primary goals of an MILP solution: minimizing the objective value and the constraint violation. The function is a

weighted sum of the objective and a penalty for constraint violations:

$$f(\boldsymbol{d}, \boldsymbol{c}) = f(\boldsymbol{x}) = \boldsymbol{w}^\top \boldsymbol{x} + \gamma \sum_{i=1}^{m} \left[ \max(0, \boldsymbol{A}_{i,*} \boldsymbol{x} - \boldsymbol{b}_i) \right]^2, \tag{5}$$

where $\boldsymbol{x} = (\boldsymbol{d}, \boldsymbol{c})$ is the set of all variables, $\gamma$ is the weight of constraint-violation term to balance two guidance parts, and $\boldsymbol{A}_{i,*}$ is the $i$-th row of the constraint matrix. The model is guided to generate solutions that minimize this function. Following Lin et al. (2025), we implement guidance by modifying the update steps for the continuous and integer variables separately.

**Guidance on Continuous Variables**. For the continuous variables, we employ gradient-based guidance. At each step $t$, we perform gradient descent on the target function $f$ w.r.t. the predicted continuous values $\hat{\boldsymbol{c}}_1(\boldsymbol{G}_t)$, steering them towards regions of lower objective value and higher feasibility:

$$\boldsymbol{c}_{t+\Delta t} \leftarrow \text{Project}_{\boldsymbol{l}, \boldsymbol{u}} \left( \boldsymbol{c}_t - \rho_t \nabla_{\boldsymbol{c}_t} f \left( \hat{\boldsymbol{d}}_{1|t}, \hat{\boldsymbol{c}}_1(\boldsymbol{G}_t) \right) \right), \tag{6}$$

where $\hat{\boldsymbol{d}}_{1|t}$ is sampled from the predicted integer distribution $\hat{p}(\boldsymbol{d}_1 | \boldsymbol{G}_t)$ and $\rho_t$ is the step size. The projection function $\text{Project}_{\boldsymbol{l}, \boldsymbol{u}}(\cdot)$ constrains the continuous variables within their specified bounds.

**Guidance on Integer Variables**. For the integer variables, we adopt an effective sampling-and-reweighting scheme motivated by recent work (Lin et al., 2025). Instead of computing gradients, at time step $t$, we sample a batch of $B$ candidate integer solutions $\{\boldsymbol{d}_{1|t,r}\}_{r=1}^{B}$ from the model's current predicted integer distribution $\hat{p}(\boldsymbol{d}_1 | \boldsymbol{G}_t)$. We then reweight the transition probabilities in the rate matrix $\hat{R}$ based on the quality of each integer candidate, as evaluated by $f(\boldsymbol{d}_{1|t,r}, \hat{\boldsymbol{c}}_1(\boldsymbol{G}_t))$, steering the categorical distribution towards more promising integer assignments:

$$\hat{R}(\boldsymbol{d}_t^{(i)}, \cdot) \leftarrow \frac{\sum_{r=1}^{B} \exp\left( f\left( \boldsymbol{d}_{1|t,r}, \hat{\boldsymbol{c}}_1(\boldsymbol{G}_t) \right) / \psi \right) \cdot R_{t|1}(\boldsymbol{d}_t^{(i)}, \cdot \mid \boldsymbol{d}_{1|t,r}^{(i)})}{\sum_{r=1}^{B} \exp\left( f\left( \boldsymbol{d}_{1|t,r}, \hat{\boldsymbol{c}}_1(\boldsymbol{G}_t) \right) / \psi \right)} \tag{7}$$

where $\psi$ is a temperature parameter. This updated rate matrix $\hat{R}$ is then used to sample the next integer state followed by the projection function for feasibility constraints, i.e., $\boldsymbol{d}_{t+\Delta t} \leftarrow \text{Project}_{\boldsymbol{l}, \boldsymbol{u}}(\boldsymbol{d}_{t+\Delta t})$.

## 4.4 Integration with Downstream Solvers

The output of the guided sampling process of FMIP is a high-quality candidate solution $(\boldsymbol{d}_1, \boldsymbol{c}_1)$ and a probability distribution over the integer variables. This provides a powerful warm-start for downstream solvers to efficiently search for better solutions. FMIP is fully compatible with a wide range of downstream solvers and can provide a powerful warm start for them to efficiently search for optimal solutions. For example, Predict-and-Search (Han et al., 2023)) employs the integer distribution to define a promising search region and uses the continuous values as an initial feasible point for the solver's LP relaxations.

## 5 Experiments

### 5.1 Experiment Settings

**Datasets & Benchmarks**   We evaluate FMIP on a comprehensive suite of eight MILP benchmarks. Five of them focus on classic combinatorial optimization problems: *Combinatorial Auctions (CA)* (Gasse et al., 2019), *Generalized Independent Set (GIS)* (Colombi et al., 2017), *Maximum Independent Set (MIS)* (Gasse et al., 2019), *Fixed-Charge Multi-Commodity Network Flow (FCMNF)* (Hewitt et al., 2010), and *Set Covering (SC)* (Gasse et al., 2019). We also include two real-world MILP datasets with both binary and continuous variables from the NeurIPS ML4CO 2021 competition (Gasse et al., 2022): *Load Balancing (LB)* and *Item Placement (IP)*. Finally, we adopt the *standard MIPLIB2017 benchmark (MIPLIB)* which is one of the most common standards for MILP solver evaluation (Gleixner et al., 2021a).

**Baselines**   We compare FMIP against three representative learning-based methods and the commercial solver Gurobi (Gurobi Optimization, LLC, 2024). The learning-based baselines are:

- **SL**: Standard supervised learning serves as a strong discriminative baseline. The model is trained to directly predict the variable assignments in a one-step manner.

- **DIFUSCO**: A state-of-the-art diffusion model for generating solutions to integer linear programs (Sun and Yang, 2023; Feng et al., 2024).

- **IP-Guided-Diff**: A guided discrete diffusion framework designed for integer programs that incorporates problem-specific guidance (Zeng et al., 2024).

Since DIFUSCO and IP-Guided-Diff are designed for integer-only problems, we adapt them to the MILP setting by including continuous variables and their constraints in the graph structure for the backbone graph encoder. In this way, we allow the baselines to perceive both the continuous and integer variables for fair comparison.

FMIP and three baselines are all learning methods that are agnostic to backbone graph neural networks and downstream solvers. In later experiments, we employ four different backbone graph encoders, i.e., Tri-GCN (Gasse et al., 2019), Bi-GCN (Gasse et al., 2019), GAT (Brody et al., 2021), and ClusterGCN (Chiang et al., 2019), with Tri-GCN as the default choice. Note that Tri-GCN and Bi-GCN both operate on a bipartite topology; the former is distinguished by handling integer and continuous variables as heterogeneous node types with distinct parameters. As for downstream solvers, we adopt Neural Diving (ND) (Nair et al., 2020), Predict-and-Search (PS) (Han et al., 2023), PMVB (Chen et al., 2025) and Apollo-MILP (Apollo) (Liu et al., 2025).

**Metrics**  Following previous works (Han et al., 2023; Liu et al., 2025), we report the *objective value* (OBJ) found by each method within a fixed time limit. To compare performance, we first determine the *best-known solution* (BKS), defined as the best objective value found across all methods, including a long run of Gurobi. We then calculate the *absolute primal gap*: $GAP = |OBJ - BKS|$. Since MIPLIB contains instances from diverse domains with very different optimal values, we report the *relative primal gap* for MIPLIB instead: $Rel.GAP = |OBJ - BKS|/(|BKS| + 1)$. A lower absolute/relative primal gap indicates better performance. Notably, we only report the Primal Gap instead of the Optimal Gap because FMIP acts as a primal heuristic. Its objective is to quickly identify high-quality feasible solutions (improving the upper bound), rather than tightening the dual bound/optimality gap. This is the standard metric for evaluating heuristic improvements.

**Implementation Details**  Due to the page limitation, we provide implementation details in Appendix D, including the backbone model architecture, training & inference hyperparameters of FMIP and baselines, as well as the downstream solver configuration.

## 5.2 OVERALL PERFORMANCE

We choose Tri-GCN as the backbone graph encoder and evaluate FMIP against baselines based on four downstream solvers (ND, PS, PMVB, and Apollo), with respective time limits of 400, 600, 600, and 800 seconds. The time limit for Gurobi is set to 3600 seconds. The results are reported in Table 1. We observe that FMIP generally achieves the best performance among learning-based methods, substantially reducing the primal gap across all benchmarks and downstream solvers. On simpler benchmarks (i.e., CA, GIS) where multiple methods find the optimal solution, FMIP performs on par with the best baselines. However, on more challenging benchmarks (i.e., LB, IP), the benefits of our joint modeling and holistic guidance are clear, culminating in the significant performance gains in terms of both objective value and primal gap. Overall, our method brings a 41.34% improvement compared with the best machine learning baseline, calculated as the mean increase over all datasets and downstream solvers.

## 5.3 COMPATIBILITY ANALYSIS

The downstream-solver compatibility of FMIP is already validated in Table 1 in Section 5.2, where FMIP consistently achieves the best performance across four different solvers. To further study the model compatibility of FMIP, we apply FMIP and baselines to four different backbone graph encoders on IP and LB benchmarks: Bi-GCN (Gasse et al., 2019), Tri-GCN (Gasse et al., 2019), GAT (Brody et al., 2021), and ClusterGCN (Chiang et al., 2019) We report the objective value (OBJ) metric in Table 2, where FMIP maintains its superior performance across all backbone graph encoders. This backbone independence demonstrates that FMIP acts as a powerful and general framework that can enhance a wide range of GNN-based models for various downstream solvers, highlighting the fundamental value of its joint distribution modeling on both continuous and integer variables.

Table 1: The overall performance of FMIP and other baselines combined with different downstream solvers. Tri-GCN is chosen as the graph encoder. Best results are given in **bold**, and the second-best values are underlined. The relative improvement is computed as $\text{Rel.Imprv.} = (\text{GAP}_{\text{best-bsl}} - \text{GAP}_{\text{FMIP}})/(\text{GAP}_{\text{best-bsl}}+1e-6)$, where $\text{GAP}_{\text{FMIP}}$ is the gap of the FMIP method, and $\text{GAP}_{\text{best-bsl}}$ is the gap of the best-performing baseline method. For MIPLIB, since we already give the relative primal gap (Rel.GAP), the relative improvement is computed as $\text{Rel.Imprv.} = (\text{Rel.GAP}_{\text{best-bsl}} - \text{Rel.GAP}_{\text{FMIP}})$

| Downstream Solver | Training Method | CA | | GIS | | MIS | | FCMNF | |
|---|---|---|---|---|---|---|---|---|---|
| | | OBJ ↑ | GAP ↓ | OBJ ↑ | GAP ↓ | OBJ ↑ | GAP ↓ | OBJ ↓ | GAP ↓ |
| ND(400s) | SL | 14691.07 | 1074.38 | 1250.10 | 573.60 | 440.30 | 9.70 | 1240317.40 | 160200.00 |
| | IP-Guided-Diff | 15132.65 | 632.80 | 1542.70 | 281.00 | 440.30 | 9.70 | 1226793.60 | 146676.20 |
| | DIFUSCO | 15042.35 | 723.10 | 1487.50 | 336.20 | 439.20 | 10.80 | 1250323.40 | 170206.00 |
| | FMIP (Ours) | 15452.75 | 312.70 | 1554.20 | 269.50 | 449.60 | 0.40 | 1221710.50 | 141593.10 |
| | Rel.Imprv. | - | 50.58% | - | 4.09% | - | 95.88% | - | 3.47% |
| PS(600s) | SL | 15765.45 | 0.00 | 1783.00 | 40.70 | 450.00 | 0.00 | 1080117.40 | 0.00 |
| | IP-Guided-Diff | 15765.45 | 0.00 | 1815.30 | 8.40 | 449.60 | 0.40 | 1080117.40 | 0.00 |
| | DIFUSCO | 15761.85 | 3.60 | 1732.60 | 91.10 | 448.50 | 1.50 | 1120110.30 | 39992.90 |
| | FMIP (Ours) | 15765.45 | 0.00 | 1816.50 | 7.20 | 450.00 | 0.00 | 1080117.40 | 0.00 |
| | Rel.Imprv. | - | 0.00% | - | 14.29% | - | 0.00% | - | 0.00% |
| PMVB(600s) | SL | 15765.45 | 0.00 | 1690.90 | 132.80 | 449.90 | 0.10 | 1080117.40 | 0.00 |
| | IP-Guided-Diff | 15765.45 | 0.00 | 1695.90 | 127.80 | 448.30 | 1.70 | 1080117.40 | 0.00 |
| | DIFUSCO | 15745.25 | 20.20 | 1673.80 | 149.90 | 443.20 | 6.80 | 1114069.00 | 33951.60 |
| | FMIP (Ours) | 15765.45 | 0.00 | 1695.90 | 127.80 | 450.00 | 0.00 | 1080117.40 | 0.00 |
| | Rel.Imprv. | - | 0.00% | - | 0.00% | - | 100.00% | - | 0.00% |
| Apollo(800s) | SL | 15765.45 | 0.00 | 1823.70 | 0.00 | 448.20 | 1.80 | 1080117.40 | 0.00 |
| | IP-Guided-Diff | 15765.45 | 0.00 | 1823.70 | 0.00 | 449.10 | 0.90 | 1080117.40 | 0.00 |
| | DIFUSCO | 15431.43 | 334.02 | 1763.24 | 60.46 | 447.50 | 2.50 | 1087517.20 | 7399.80 |
| | FMIP (Ours) | 15765.45 | 0.00 | 1823.70 | 0.00 | 450.00 | 0.00 | 1080117.40 | 0.00 |
| | Rel.Imprv. | - | 0.00% | - | 0.00% | - | 100.00% | - | 0.00% |
| Gurobi(3600s) | — | 15765.45 | 0.00 | 1823.70 | 0.00 | 450.00 | 0.00 | 1080117.40 | 0.00 |

| Downstream Solver | Training Method | SC | | LB | | IP | | MIPLIB | |
|---|---|---|---|---|---|---|---|---|---|
| | | OBJ ↓ | GAP ↓ | OBJ ↓ | GAP ↓ | OBJ ↓ | GAP ↓ | OBJ ↓ | Rel.GAP ↓ |
| ND(400s) | SL | 401.00 | 0.30 | 719.67 | 13.77 | 14.62 | 0.88 | 1619541.12 | 6.21% |
| | IP-Guided-Diff | 400.95 | 0.25 | 712.34 | 6.44 | 14.63 | 0.89 | 1609725.33 | 5.85% |
| | DIFUSCO | 402.10 | 1.40 | 717.73 | 11.83 | 15.41 | 1.67 | 1625188.49 | 6.48% |
| | FMIP (Ours) | 400.80 | 0.10 | 706.30 | 0.40 | 13.99 | 0.25 | 1604917.80 | 5.42% |
| | Rel.Imprv. | - | 60.00% | - | 93.79% | - | 71.59% | - | 0.43% |
| PS(600s) | SL | 400.80 | 0.00 | 749.60 | 43.70 | 15.34 | 1.60 | 1612105.77 | 5.67% |
| | IP-Guided-Diff | 400.75 | 0.05 | 725.34 | 19.44 | 15.21 | 1.47 | 1601226.94 | 5.31% |
| | DIFUSCO | 401.30 | 0.60 | 714.11 | 8.21 | 14.73 | 0.99 | 1611839.42 | 5.65% |
| | FMIP (Ours) | 400.70 | 0.00 | 705.90 | 0.00 | 13.92 | 0.18 | 1595308.15 | 4.84% |
| | Rel.Imprv. | - | 100.00% | - | 100.00% | - | 74.86% | - | 0.47% |
| PMVB(600s) | SL | 422.10 | 21.40 | 706.10 | 0.20 | 15.39 | 1.65 | 1620951.88 | 6.34% |
| | IP-Guided-Diff | 405.50 | 4.80 | 706.30 | 0.40 | 15.17 | 1.43 | 1614667.01 | 5.88% |
| | DIFUSCO | 412.60 | 11.90 | 714.21 | 8.31 | 15.03 | 1.29 | 1628433.01 | 6.62% |
| | FMIP (Ours) | 400.80 | 0.10 | 706.10 | 0.20 | 14.41 | 0.67 | 1608125.64 | 5.55% |
| | Rel.Imprv. | - | 97.92% | - | 0.00% | - | 48.06% | - | 0.57% |
| Apollo(800s) | SL | 400.80 | 0.10 | 706.00 | 0.10 | 14.16 | 0.42 | 1604557.34 | 4.98% |
| | IP-Guided-Diff | 405.50 | 4.80 | 705.95 | 0.05 | 14.33 | 0.59 | 1593821.50 | 4.55% |
| | DIFUSCO | 403.30 | 2.60 | 709.32 | 3.42 | 14.01 | 0.27 | 1591488.19 | 4.43% |
| | FMIP (Ours) | 400.70 | 0.00 | 705.90 | 0.00 | 13.74 | 0.00 | 1586142.71 | 4.15% |
| | Rel.Imprv. | - | 100.00% | - | 100.00% | - | 100.00% | - | 0.28% |
| Gurobi(3600s) | — | 400.80 | 0.10 | 706.00 | 0.10 | 17.73 | 3.99 | 1522661.56 | 0.00% |

## 5.4 INFERENCE EFFICIENCY

We report the averaged inference time of FMIP and baselines to generate a heuristic solution for one MILP problem on different benchmarks. Tri-GCN is adopted as the default backbone graph encoder. Note that most downstream solvers (ND, PS, and PMVB) only need to obtain the heuristic solution based on one inference process from the model. However, Apollo is a special solver that requires the model to infer multiple times for each single MILP instance, with the problem size gradually reduced and the solution iteratively refined. In this experiment, we iterate 3 times for Apollo.

As shown in Table 3, we report the inference time per MILP instance separately for ND/PS/PMVB and Apollo. Despite the superior performance of FMIP, its generative process is also highly efficient.

Table 2: The model compatibility study, where we apply FMIP and baselines to different backbone graph neural networks. We report the objective value (OBJ) metric. The best results are given in **bold**, and the second-best values are underlined.

| Downstream Solver | Training Method | Item Placement (IP) | | | | Load Balancing (LB) | | | |
|---|---|---|---|---|---|---|---|---|---|
| | | Bi-GCN | Tri-GCN | GAT | ClusterGCN | Bi-GCN | Tri-GCN | GAT | ClusterGCN |
| ND(400s) | SL | 14.85 | 14.62 | 14.65 | 14.69 | 719.54 | 719.67 | 719.20 | 720.15 |
| | IP-Guided-Diff | 14.82 | 14.63 | 14.63 | 14.63 | 713.32 | 712.34 | 712.20 | 713.10 |
| | DIFUSCO | 15.44 | 15.41 | 15.39 | 15.50 | 717.75 | 717.73 | 717.68 | 717.75 |
| | FMIP | **14.11** | **13.99** | **13.95** | **14.06** | **707.24** | **706.30** | **706.50** | **706.50** |
| PS(600s) | SL | 15.38 | 15.34 | 15.33 | 15.32 | 750.34 | 749.60 | 749.26 | 749.69 |
| | IP-Guided-Diff | 15.24 | 15.21 | 15.21 | 15.21 | 726.04 | 725.34 | 725.46 | 725.72 |
| | DIFUSCO | 14.73 | 14.73 | 14.76 | 14.81 | 714.01 | 714.11 | 714.77 | 714.90 |
| | FMIP | **13.92** | **13.92** | **13.94** | **13.92** | **705.88** | **705.90** | **706.10** | **706.00** |
| PMVB(600s) | SL | 15.36 | 15.39 | 15.36 | 15.44 | 706.50 | 706.10 | 706.30 | 706.30 |
| | IP-Guided-Diff | 15.25 | 15.17 | 15.15 | 15.17 | 706.62 | 706.30 | 706.20 | 706.30 |
| | DIFUSCO | 15.07 | 15.03 | 15.03 | 15.09 | 714.89 | 714.21 | 714.01 | 714.55 |
| | FMIP | **14.45** | **14.41** | **14.42** | **14.39** | **706.32** | **706.10** | **706.10** | **706.20** |
| Apollo(800s) | SL | 14.19 | 14.16 | 14.11 | 14.19 | 706.30 | 706.25 | 706.20 | 706.30 |
| | IP-Guided-Diff | 14.36 | 14.03 | 14.33 | 14.40 | 706.00 | 705.95 | 706.10 | 706.10 |
| | DIFUSCO | 14.01 | 14.01 | 13.99 | 14.05 | 709.92 | 709.32 | 709.40 | 709.40 |
| | FMIP | **13.72** | **13.74** | **13.74** | **13.75** | **705.90** | **705.90** | **705.90** | **706.00** |

Table 3: The inference time (s) per MILP instance of FMIP and baselines. We only compare the neural model inference here, including the computation of guidance, and exclude the time that downstream solvers take.

| Downstream Solver | Training Method | CA | GIS | MIS | FCMNF | SC | LB | IP | MIPLIB |
|---|---|---|---|---|---|---|---|---|---|
| ND(400s), PS(600s), PMVB(600s) | SL | 0.047 | 0.089 | 0.138 | 0.065 | 0.087 | 0.088 | 0.023 | 0.115 |
| | DIFUSCO | 0.187 | 0.652 | 0.340 | 0.238 | 0.412 | 1.154 | 0.093 | 1.561 |
| | IP-Guided-Diff | 0.213 | 0.712 | 0.452 | 0.374 | 0.533 | 1.781 | 0.117 | 2.153 |
| | FMIP | 0.281 | 0.761 | 0.476 | 0.310 | 0.623 | 1.294 | 0.176 | 2.043 |
| Apollo(800s) | SL | 0.082 | 0.273 | 0.184 | 0.310 | 0.209 | 0.213 | 0.093 | 0.327 |
| | DIFUSCO | 0.512 | 2.125 | 1.341 | 0.545 | 1.112 | 3.129 | 0.153 | 3.231 |
| | IP-Guided-Diff | 0.612 | 2.173 | 1.549 | 0.634 | 1.268 | 3.151 | 0.167 | 4.185 |
| | FMIP | 0.718 | 2.060 | 1.805 | 0.546 | 1.782 | 3.229 | 0.121 | 4.812 |

The inference time of FMIP is comparable to that of other generative baselines (i.e., DIFUSCO and IP-Guided-Diff). Crucially, this time represents a negligible fraction (often less than 1%) of the total time spent by the downstream solver, which can be hundreds or thousands of seconds. FMIP thus provides its substantial performance improvements with minimal computational overhead, striking an excellent balance between solution quality and efficiency.

## 5.5 IN-DEPTH ANALYSIS

**Ablation Study**. We derive four variants of our proposed FMIP: (i) removing the objective guidance in Eq. 5 (*w/o Feasibility Guidance*), (ii) removing the objective guidance in Eq. 5 (*w/o Objective Guidance*), (iii) removing the entire holistic guidance mechanism(*w/o Guidance*), and (iv) reverting to an integer-only model by disabling continuous variable generation (*w/o Continuous*), which inherently prevents the application of holistic guidance. As shown in Table 4, the full version of FMIP significantly outperforms all ablated variants, confirming that both the joint generation of all variables and the holistic guidance mechanism are critical to the performance gain. Notably, the "w/o Continuous" variant suffers from the worst performance, providing direct evidence that capturing the coupled relationship between integer and continuous variables is essential for high-quality solution prediction in MILP.

**Impact of Sampling Steps**. We analyze the effect of the number of sampling steps on the heuristic solution quality. Specifically, we feed the heuristic solutions from different sampling steps into the downstream solvers, and report the primal gap metric (i.e., GAP) in Figure 3. We can observe that, at the beginning, more sampling steps lead to better solutions as the model has more opportunities for refinement. However, the quality of heuristic solution starts to worsen when a plethora of steps are performed. We attribute this phenomenon to the trade-off between generative refinement and search diversity. An excessive number of sampling steps might cause the guidance to make the predicted variable distribution overly sharp, which reduces the diversity of the solutions being explored and can prematurely trap the search in a local optimum. Alternatively, this degradation may stem from

Table 4: Ablation study of FMIP and its variants. The best results are given in **bold**, and the second-best values are underlined.

| Variant | ND | | PS | | PMVB | | Apollo | |
|---|---|---|---|---|---|---|---|---|
| | **OBJ**↓ | **GAP**↓ | **OBJ**↓ | **GAP**↓ | **OBJ**↓ | **GAP**↓ | **OBJ**↓ | **GAP**↓ |
| **Full FMIP** | **13.99** | **0.25** | **13.92** | **0.18** | **14.41** | **0.67** | **13.74** | **0.00** |
| w/o Feasibility Guidance | 14.75 | 1.01 | 14.67 | 0.93 | 15.47 | 1.73 | 14.97 | 1.23 |
| w/o Objective Guidance | 14.65 | 0.91 | 14.76 | 1.02 | 22.32 | 8.58 | 14.53 | 0.79 |
| w/o Guidance | 14.58 | 0.84 | 13.98 | 0.24 | 14.45 | 0.71 | 14.11 | 0.37 |
| w/o Continuous | 14.43 | 0.69 | 14.28 | 0.54 | 14.95 | 1.21 | 14.45 | 0.71 |

the balance between objective value and feasibility in the target function 5, which is discussed in the subsequent part.

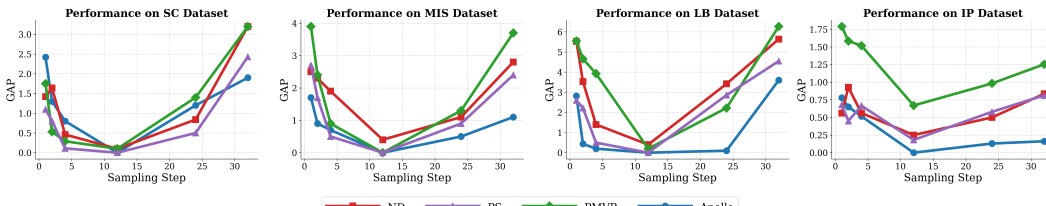

Figure 3: The performance of FMIP with downstream solvers w.r.t. different sampling steps. We report the absolute primal gap (GAP) as the metric.

**Adaptive Guidance Balancing**. We investigate the potential imbalance between objective and feasibility guidance terms during iterative generation. We hypothesize that a fixed coefficient $\gamma$ (defaulting to 50) leads to performance degradation over longer sampling trajectories. To address this, we introduce an adaptive balancing strategy that dynamically adjusts $\gamma$ at each generation step. Specifically, we compare the magnitude of the objective guidance term with that of the constraint-violation term: if the objective term dominates, we scale up $\gamma$ by a factor $\alpha$ (i.e., $\gamma \leftarrow \gamma \times \alpha$) to enforce stronger constraint satisfaction; conversely, if the constraint term prevails, we scale down $\gamma$ (i.e., $\gamma \leftarrow \gamma/\alpha$). This adjustment is cumulative across sampling steps, allowing the model to dynamically recalibrate the guidance focus between optimality and feasibility. We evaluate this strategy on the Item Placement dataset using the PS solver, varying $\alpha$ and the number of steps. As shown in Table 5, with a fixed $\gamma$, the objective value worsens significantly as steps increase (e.g., from 11.79 at Step-12 to 13.25 at Step-32). However, the adaptive strategy ($\alpha > 1$) effectively mitigates this degradation, maintaining competitive performance even at deeper generation steps (e.g., 11.92 at Step-32 with $\alpha = 2.0$). These results validate that dynamically balancing guidance is crucial for stabilizing long-trajectory generation.

Table 5: Impact of Adaptive Balancing Strategy on Objective Value (Lower is Better) across different sampling steps on the IP dataset.

| Strategy | Step-12 | Step-24 | Step-32 |
|---|---|---|---|
| Fixed $\gamma$ | **11.79** | 12.39 | 13.25 |
| Adaptive ($\alpha = 1.1$) | **11.93** | 12.35 | 12.05 |
| Adaptive ($\alpha = 1.5$) | 11.99 | **11.85** | 12.68 |
| Adaptive ($\alpha = 2.0$) | 12.19 | 11.95 | **11.92** |

## 6 CONCLUSION

In this paper, we propose Joint Continuous-Integer Flow for Mixed Integer Linear Programming (i.e., **FMIP**), which is the first generative framework to model the *joint distribution* of both integer and continuous variables. Based on this, we further design the *holistic guidance mechanism* that uses instance-specific objective and constraint information to steer the generative process toward higher-quality heuristic solutions. As a powerful generative learning framework, FMIP is *fully compatible* with arbitrary backbone graph encoders and various downstream solvers, demonstrating its great potential for real-world MILP applications. Extensive experiments show that FMIP sets new state-of-the-art performance for learning-based MILP heuristics, reducing the primal gap by 41.34% on average compared to the best baseline method. Two key directions for future work are: enhancing the graph representation to better capture task-specific features, and developing a customized solver tailored to our FMIP framework to optimize solution quality and computational efficiency.

ACKNOWLEDGMENT

Lin's research is partially supported by the National Natural Science Foundation of China (NSFC) [Grant NSFC-624B2096, 72542012, 72595872]. Ge's research is partially supported by the National Natural Science Foundation of China (NSFC) [Grant NSFC-72225009, 72394360, 72394365].

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

## A  LLM USAGE DECLARATION

In this submission, we used a Large Language Model (LLM) solely for language refinement and text polishing. The LLM was employed to enhance the clarity, flow, and readability of the manuscript, but it did not contribute to the ideation, analysis, or generation of scientific content. All ideas, interpretations, and results presented in the paper are solely the work of the authors. No content generated by the LLM was used to fabricate facts or contribute to the research findings.

## B  GRAPH REPRESENTATION DETAILS

Building on previous work, we select a similar graph structure. To more naturally accommodates our joint generation purpose, we explicitly distinguishing three types of nodes: Integer Variables (Ivar), Continuous Variables (Cvar), and Constraints (con). Edges, denoted as $e_{ij}$, are drawn between the variable nodes (both Ivar and Cvar) and the constraint nodes to capture their algebraic relationships within the MILP formulation. The edge weights $e_{ij}$ correspond to the coefficients of variables in their associated constraints. The features of nodes and edges of a MILP graph $\boldsymbol{G} = (\mathcal{V}_{\text{Ivar}}, \mathcal{V}_{\text{Cvar}}, \mathcal{V}_{\text{con}}, \mathcal{E})$ are:

- **Ivar/Cvar Nodes** ($\mathcal{V}_{\text{Ivar}}/\mathcal{V}_{\text{Cvar}}$): The feature vector for variable $v_i \in \mathcal{V}_{\text{Ivar}} \cup \mathcal{V}_{\text{Cvar}}$ is defined as:

$$\boldsymbol{h}_{v_i} = \left[ \underbrace{\boldsymbol{w}_i}_{\text{objective coefficient}}, \underbrace{\boldsymbol{l}_i}_{\text{lower bound}}, \underbrace{\boldsymbol{u}_i}_{\text{upper bound}}, \underbrace{\boldsymbol{hlb}_i}_{\text{if } l_i > -\infty}, \underbrace{\boldsymbol{hub}_i}_{\text{if } u_i < +\infty} \right] \in \mathbb{R}^5,$$

  where $\boldsymbol{w}, \boldsymbol{l}, \boldsymbol{u}$ are defined in Eq. 1.

- **Constraint Nodes** ($\mathcal{V}_{\text{con}}$): The feature for constraint $con_j$ is scalar $b_j \in \mathbb{R}$, representing its right-hand side constant.

- **Edges** ($\mathcal{E}$): We construct a sparse bipartite graph based on the nonzero entries of the constraint matrix $\mathbf{A}$. Specifically, an edge is added between variable node $j$ and constraint node $i$ if and only if $\mathbf{A}[i,j] \neq 0$. The edge feature is set as the corresponding coefficient, i.e., the edge from $j$ to $i$ carries weight $\mathbf{A}[i,j]$.

When processing with a graph neural network, the state in generative model $\boldsymbol{x}_t = (\boldsymbol{d}_t, \boldsymbol{c}_t)$ is incorporated into the features of the corresponding variable nodes in $\boldsymbol{G}$. Specifically, for each variable node $v_i \in \mathcal{V}_{\text{Ivar}} \cup \mathcal{V}_{\text{Cvar}}$, its static feature vector $\boldsymbol{h}_{v_i}$ from the MILP graph is augmented with its current value from the solution vector $\boldsymbol{x}_t$, forming the time-dependent feature vector $\boldsymbol{h}_{v_i,t} = [\boldsymbol{h}_{v_i}, \boldsymbol{x}_t^{(i)}] \in \mathbb{R}^6$. This implies that the static variable features $\boldsymbol{h}_{v_i}$ are 5-dimensional. Constraint node features are typically static. Figure 4 provides a simple example illustrating the structure of a solution graph derived from an MILP instance.

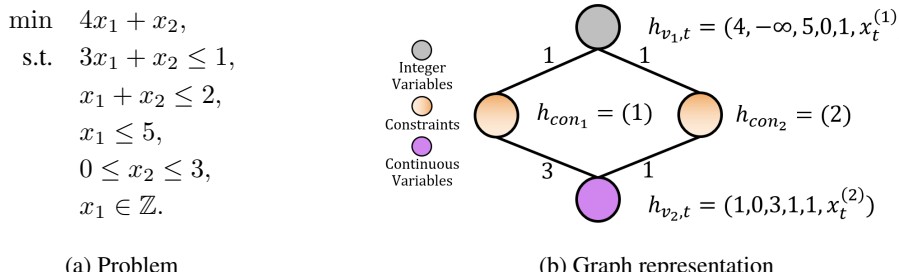

$$
\begin{aligned}
\min \quad & 4x_1 + x_2, \\
\text{s.t.} \quad & 3x_1 + x_2 \leq 1, \\
& x_1 + x_2 \leq 2, \\
& x_1 \leq 5, \\
& 0 \leq x_2 \leq 3, \\
& x_1 \in \mathbb{Z}.
\end{aligned}
$$

(a) Problem

(b) Graph representation

Figure 4: Toy example of a MILP instance and its graph representation in Flow Matching.

## C  GRAPH NEURAL NETWORKS

The input to our neural network is a tuple $(\boldsymbol{G}_t, t)$, where $\boldsymbol{G}_t$ represents the solution graph as described in Section B. This graph encodes the relationships between variables and constraints, effectively capturing the structure of the MILP problem. The scalars $t$ are temporal parameters associated with the integer and continuous variables, respectively, with both $t \in [0, 1]$. These time parameters are introduced to enable a dynamic, progressive refinement of the solution space during the training process, aligning with the flow matching paradigm where time plays a critical role in guiding the

evolution of the generated solution. We'll details our neural networks structure in the following subsection.

## C.1 GCNConv Layer

We adopt the following mathematical notations for the following parts: $\text{Linear}_{.}(\cdot)$ denotes a learnable affine transformation (linear layer), $\text{LN}_{.}(\cdot)$ represents layer normalization, and $\text{GELU}(\cdot)$ indicates Gaussian Error Linear Unit activation. The element-wise sigmoid function is written as $\sigma(\cdot)$, while $\odot$ signifies element-wise multiplication. Vector concatenation is denoted by $[\cdot\,;\,\cdot]$, and $\mathbb{I}_{\text{res}} \in \{0, 1\}$ serves as an indicator variable for residual connection usage.

The output feature $h'_v$ of a target node $v$ is computed as:

$$h'_v = \text{LN}_{\text{out}}\left(\text{Linear}_{o2}\left(\text{GELU}\left(\text{Linear}_{o1}\left([\text{LN}_{\text{PC}}(g_v \odot a_1 + (1 - g_v) \odot a_2)\,;\,h_v]\right)\right)\right)\right) + \mathbb{I}_{\text{res}} \cdot h_v,$$

where the aggregated messages from source node types $k = 1, 2$ are defined as:

$$a_k = \sum_{u \in \mathcal{N}_k(v)} \text{Linear}_{\text{final}}\left(\text{GELU}\left(\text{LN}_{\text{PN}}\left(\text{Linear}_t(h_v) + \text{Linear}_{s_k}(h_u^{(k)}) + \text{Linear}_e(e_{uv}^{(k)})\right)\right)\right),$$

and the gating vector is computed by: $g_v = \sigma\left(\text{Linear}_{fg}([a_1; a_2])\right)$.

## C.2 Model Structure

**Overview** We propose a graph-based neural architecture tailored for mixed-variable optimization problems. The model captures the interactions among integer variables, continuous variables, and constraints using iterative message passing with temporal conditioning.

**Initial Embeddings** We define three types of nodes:

- Integer variables ($\mathcal{V}_{\text{Ivar}}$): $n_{\text{Ivar}}$ nodes, with features $\mathbf{X}_{\text{Ivar}} \in \mathbb{R}^{n_{\text{Ivar}} \times d_{\text{Ivar}}}$
- Continuous variables ($\mathcal{V}_{\text{Cvar}}$): $n_{\text{Cvar}}$ nodes, with features $\mathbf{X}_{\text{Cvar}} \in \mathbb{R}^{n_{\text{Cvar}} \times d_{\text{Cvar}}}$
- Constraints ($\mathcal{C}_{\text{con}}$): $n_{\text{con}}$ nodes, with features $\mathbf{X}_{\text{con}} \in \mathbb{R}^{n_{\text{con}} \times d_{\text{con}}}$

Initial node embeddings are computed via type-specific MLPs:

$$\mathbf{h}_{\text{Ivar}}^{(0)} = \text{MLP}_{\text{Ivar}}(\mathbf{X}_{\text{Ivar}}), \quad \mathbf{h}_{\text{Cvar}}^{(0)} = \text{MLP}_{\text{Cvar}}(\mathbf{X}_{\text{Cvar}}), \quad \mathbf{h}_{\text{con}}^{(0)} = \text{MLP}_{\text{con}}(\mathbf{X}_{\text{con}}),$$

where each embedding lies in $\mathbb{R}^{\cdot \times h}$ with $h$ denoting the shared hidden dimension.

The time embedding $\mathbf{e}_t \in \mathbb{R}^h$ is initialized using positional encoding(Vaswani et al., 2017). This embedding is broadcasted and added to the variable node features at every iteration.

**Message Passing Layers** For each layer $\ell = 1, \ldots, L$, we update the node representations via a message passing scheme over a bipartite graph with three node types. The relevant edge sets are defined as:

- $\mathcal{E}_{\text{Ivar2con}} \subseteq \mathcal{V}_{\text{Ivar}} \times \mathcal{C}_{\text{con}}$: integer-to-constraint edges
- $\mathcal{E}_{\text{Cvar2con}} \subseteq \mathcal{V}_{\text{Cvar}} \times \mathcal{C}_{\text{con}}$: continuous-to-constraint edges
- Reverse edges $\mathcal{E}_{\text{con2Ivar}}, \mathcal{E}_{\text{con2Cvar}}$: constraint-to-variable edges for backward message flow

The node updates at layer $\ell$ are:

$$\mathbf{h}_t^{(\ell)} = \text{MLP}_t^{(\ell)}(\mathbf{e}_t),$$
$$\mathbf{h}_{\text{con}}^{(\ell)} = \mathbf{h}_{\text{con}}^{(\ell-1)} + \mathbf{h}_t^{(\ell)} + \text{MLP}_{\text{con}}^{(\ell)}\left(\text{TriConv}\left(\mathbf{h}_{\text{Ivar}}^{(\ell-1)}, \mathbf{h}_{\text{Cvar}}^{(\ell-1)}, \mathbf{h}_{\text{con}}^{(\ell-1)}, \mathcal{E}_{\text{Ivar2con}}, \mathcal{E}_{\text{Cvar2con}}\right)\right),$$
$$\mathbf{h}_*^{(\ell)} = \mathbf{h}_*^{(\ell-1)} + \mathbf{h}_t^{(\ell)} + \text{MLP}_*^{(\ell)}\left(\text{BiConv}_*(\mathbf{h}_{\text{con}}^{(\ell)}, \mathbf{h}_*^{(\ell-1)}, \mathcal{E}_{\text{con2}*})\right), \quad * \in \{\text{Ivar}, \text{Cvar}\}.$$

`TriConv` is a message aggregation module that aggregates signals from two types of source nodes into target nodes. `BiConv` has the same architecture as `TriConv`, but aggregates signals from one type of source nodes into target nodes.

**Output Layer** After $L$ layers of message passing, we apply type-specific MLP heads to predict outputs from the final node embeddings:

$$\mathbf{o}_{\text{Ivar}} = \text{MLP}_{\text{out}}^{\text{Ivar}}(\mathbf{h}_{\text{Ivar}}^{(L)}), \quad \mathbf{o}_{\text{Cvar}} = \text{MLP}_{\text{out}}^{\text{Cvar}}(\mathbf{h}_{\text{Cvar}}^{(L)}),$$

where $\mathbf{o}_{\text{Ivar}} \in \mathbb{R}^{n_{\text{Ivar}} \times d_{\text{out}}^{\text{Ivar}}}$ and $\mathbf{o}_{\text{Cvar}} \in \mathbb{R}^{n_{\text{Cvar}} \times d_{\text{out}}^{\text{Cvar}}}$ are the prediction outputs for integer and continuous variables, respectively.

# D  IMPLEMENTATION DETAILS

We'll present the implementation details in this section, which contains infrastructure and hyperparameter in our proposed methods.

## D.1  TRAINING DETAILS

Training labels for FMIP and other baselines were generated by solving the MILP instances using COPT (Ge et al., 2024), an outperforming MILP solver (Gleixner et al., 2021b), with a time limit of 3600 seconds per instance and 12 threads. The training loss for the GNN baseline is binary cross entropy for the prediction of binary variables, and the training loss for FMIP is described in Eq. 4. For MIPLIB, the training set is collected from all other benchmarks due to its diverse problem types. For the other datasets, the training set is split from the entire data with a 9:1 ratio.

## D.2  INFRASTRUCTURE

Our flow model is implemented in PyTorch(Paszke et al., 2019) and PyTorch Geometric(Fey and Lenssen, 2019) and trained on a single NVIDIA H100 GPU.For CPU, we we use 12 cores of an Intel Xeon Platinum 8469C at 2.60 GHz CPU with 512 GB RAM. We select Gurobi(Gurobi Optimization, LLC, 2024) , which is the most famous and well-implement MILP solver, in Decoding Strategy and baseline methods due to its high efficiency. For all experiments, we set 12 threads for backend solver, Gurobi, which is a standard setting (Gleixner et al., 2021b).

Table 6: Experimental Parameters Comparison

| **Method** | **ND** | **PS** | **PMVB** | **Apollo** |
|---|---|---|---|---|
| (Parameters) | $[K, \alpha]$ | $[k_0, k_1, \Delta]$ | $[\delta, \tau]$ | $[k_0, k_1, \Delta, K]$ |
| Cauctions | $[50, 0.1]$ | $[0.3, 0.06, 0.3]$ | $[0.7, 0.9]$ | $[0.3, 0.06, 0.3, 2]$ |
| GISP | $[50, 0.1]$ | $[0.2, 0.02, 0.2]$ | $[0.7, 0.9]$ | $[0.2, 0.02, 0.2, 2]$ |
| Independent Set | $[50, 0.1]$ | $[0.3, 0.2, 0.3]$ | $[0.7, 0.9]$ | $[0.3, 0.2, 0.3, 2]$ |
| FCMNF | $[50, 0.1]$ | $[0.3, 0.03, 0.2]$ | $[0.7, 0.9]$ | $[0.3, 0.03, 0.2, 2]$ |
| Item Placement | $[50, 0.1]$ | $[0.3, 0.08, 0.4]$ | $[0.7, 0.9]$ | $[0.3, 0.08, 0.4, 2]$ |
| Load Balancing | $[50, 0.1]$ | $[0.2, 0.2, 0.2]$ | $[0.7, 0.9]$ | $[0.2, 0.2, 0.2, 2]$ |
| Set Covering | $[50, 0.1]$ | $[0.3, 0.04, 0.2]$ | $[0.7, 0.9]$ | $[0.3, 0.04, 0.2, 2]$ |

## D.3  DOWNSTREAM SOLVERS

Table 6 summarizes the hyperparameter settings used by each downstream solver across different problem instances. The meaning of each parameter (e.g., $K$, $\alpha$, $k_0$, $\Delta$, etc.) is consistent with the notation defined in Appendix E, where detailed descriptions of downstream solvers and their hyperparameters are provided. For a fair comparison, we use the same parameters for the downstream solvers when evaluating FMIP and all baselines.

## D.4  FMIP

This section outlines the key hyperparameters employed during model training and inference, covering: (1) general training configurations (e.g., learning rate, batch size), (2) neural network architecture specifications, and (3) inference-specific settings such as sampling strategies and guidance control. The fixed default parameters are provided in Table 7, while two critical parameters, Batch Size and Guidance Temperature, are dynamically adjusted based on dataset characteristics, detailed in following parts.

**Choice of Batch Size**    In practice, Batch size is dynamically adjusted according to the problem scale and available GPU memory. We adopt the largest feasible batch size that fits into memory to maximize hardware utilization and training efficiency.

Table 7: Training and Inference Configuration for FMIP

| **Parameter Description** | **Value** |
|---|---|
| Training epochs | 300 |
| Learning rate | 2e-4 |
| Weight decay | 1e-4 |
| Learning rate scheduler | cosine-decay |
| GNN layers | 12 |
| Hidden dimension | 64 |
| Inference schedule | cosine |
| Inference Steps | 12 |
| Integer guidance temperature ($\psi$) | 0.1, 1, 10 |
| Continuous guidance stepsize ($\rho$) | 0.1 |
| Weight of constraint violation ($\gamma$) | 50 |

**Guidance Temperature** We temperature coeffient ($\psi$) is introduced to control the strength of guidance during inference, where lower temperature represents higher strength. This parameter is selected based on validation performance. Specifically, we set the temperature to 0.1 for the problem Load Balancing and Item Placement, 10 for the Set Covering problem, and 1 for all other problem types.

# E  DOWNSTREAM SOLVER ALGORITHM

In this section, we introduction detailed algorithms employed in our experiments, including Neural Diving, Predict-and-Search, PMVB and Apollo-MILP.

## E.1  NEURAL DIVING

For a new instance, the neural network first generates multiple partial variable assignments based on predicted probability distributions. Let $\mathcal{V}$ denote the set of variables. The algorithm fixes a subset $\mathcal{S} \subset \mathcal{V}$ where:

$$|\mathcal{S}| = \alpha \cdot |\mathcal{V}| \quad (\alpha \in (0,1))$$

and generates $K$ parallel sub-MIPs through $\{\mathcal{S}_i\}_{i=1}^{K}$ assignments. Through a selective prediction mechanism, it decides which variables to fix and samples their specific values, forming various assignment combinations that only involve subsets of variables. Subsequently, each partial assignment is transformed into a smaller sub-MIP problem by fixing these assigned variables. These sub-MIP problems are then solved in parallel using traditional solvers to generate multiple complete feasible solutions. Finally, the solution with the optimal objective function value is selected as the final result. This process combines the predictive capabilities of neural networks with the efficiency of traditional solvers, achieving an end-to-end workflow from partial assignments to optimal solutions.

## E.2  PREDICT-AND-SEARCH

In the predict phase, a graph neural network estimates binary variable assignments $\hat{x}_j \in \{0, 1\}$. During the search phase, the algorithm constructs a trust region based on the predictions, restricting the original problem to a feasible solution space within the neighborhood of the predicted solutions. The trust region is defined as follows:

$$\mathcal{T}_0 = \{x_j \mid \hat{x}_j \leq k_0\} \quad \text{(variables near 0)}$$
$$\mathcal{T}_1 = \{x_j \mid \hat{x}_j \geq 1 - k_1\} \quad \text{(variables near 1)}$$
$$\sum_{j \in \mathcal{T}_0} \mathbb{I}(x_j^* = 1) + \sum_{j \in \mathcal{T}_1} \mathbb{I}(x_j^* = 0) \leq \Delta \cdot (|\mathcal{T}_0| + |\mathcal{T}_1|)$$

where $k_0, k_1 \in (0, 1)$ control selection thresholds and $\delta$ defines permissible deviation. Specifically, by adding constraints (e.g., limiting the number of variables with scores close to 0 or 1 and controlling deviations from predicted values), the problem is transformed into a smaller subproblem, which is then solved using traditional solvers to efficiently search for high-quality feasible solutions. Compared to methods that directly fix variables, the trust region strategy retains flexibility to avoid infeasibility while improving solution quality through localized search.

## E.3  PMVB

The PMVB (Probabilistic Multi-Variable Cardinality Branching) method leverages predictions from a graph neural network to construct statistically grounded branching constraints. Specifically, the binary variables are partitioned into two disjoint sets based on their predicted probabilities:

$$\mathcal{U} = \{j : \hat{y}_j \geq \tau\}, \quad \mathcal{L} = \{j : \hat{y}_j \leq 1 - \tau\}$$

where $\tau \in (0.5, 1]$ is a confidence threshold indicating how certain the model is about a variable being 1 or 0, respectively. Using the principles of risk pooling and concentration inequalities, the algorithm constructs two soft constraints that serve as branching hyperplanes:

$$\mathcal{C}_{\mathcal{U}} : \sum_{j \in \mathcal{U}} y_j \geq \left\lceil (1 - \delta) \sum_{j \in \mathcal{U}} \mathbb{E}[\hat{y}_j] - \Gamma \right\rceil, \quad \mathcal{C}_{\mathcal{L}} : \sum_{j \in \mathcal{L}} y_j \leq \left\lfloor \delta \sum_{j \in \mathcal{L}} \mathbb{E}[\hat{y}_j] + \Gamma \right\rfloor$$

where $\Gamma = \sqrt{\frac{|\mathcal{S}| \ln(2/\delta)}{2}}$ for $\mathcal{S} \in \{\mathcal{U}, \mathcal{L}\}$, and $\delta$ is a tunable confidence parameter.

These hyperplanes probabilistically constrain the variable assignments, forming a high-confidence trust region that filters out unlikely configurations while preserving feasible and high-quality candidates. By intersecting the feasible region with both $\mathcal{C}_{\mathcal{U}}$ and $\mathcal{C}_{\mathcal{L}}$, the algorithm defines a subproblem $\mathcal{P}_{\mathcal{C}_{\mathcal{U}}, \mathcal{C}_{\mathcal{L}}}$ that is both reduced in size and refined in quality.

This subproblem is then passed to a traditional MILP solver for efficient resolution. Compared to methods that directly fix variables, PMVB offers greater robustness by retaining feasible flexibility while introducing statistically grounded directional guidance. As a result, it effectively balances confidence-driven pruning with solution diversity, improving both computational efficiency and solution quality in challenging problem instances.

### E.4 APOLLO-MILP

The algorithm begins by employing a graph neural network to predict values for currently unfixed variables, yielding a predicted solution. Subsequently, the Predict-and-Search method is applied to solve subproblems derived from this prediction, generating a reference solution. By comparing the predicted and reference solutions, variables with identical values in both solutions are fixed, thereby constructing a reduced problem. At each iteration $t \in \{1, ..., K\}$, the algorithm applies parameters $[k_0^{(t)}, k_1^{(t)}, \Delta^{(t)}]$ in Predict-and-Search algorithm, where $K$ controls total iterations. This process iteratively repeats, progressively identifying high-quality partial solutions and expanding the subset of fixed variables to reduce problem dimensionality.

Throughout the iterations, Apollo-MILP alternates between prediction and correction steps, continuously refining the predicted solution and reducing the complexity of the MILP problem. This approach ensures optimality while significantly enhancing both solution quality and computational efficiency.

## F   PSEUDO CODE FOR FMIP INFERENCE

---

**Algorithm 1** Inference Phase of FMIP

---

**Require:**

  Rectified flow model $g_\theta$

  Target function $f$, guidance strength $\rho$, temperature $\tau$

  Number of guidance steps $N_{\text{iter}}$

  Number of inference steps $N_{\text{in}}$, temporal step size $\Delta t = [\Delta t_1 = 0, \Delta t_2, \cdots, \Delta t_{N_{\text{in}}} = 1]$

1: **procedure** MAIN
2:     Convert an MILP instance to Graph $\boldsymbol{G}$
3:     Get $\boldsymbol{l}, \boldsymbol{u} \in \mathbb{R}^n$ for constraints $\boldsymbol{l} \le \boldsymbol{x} \le \boldsymbol{u}$, and let $\mathcal{B}$ represent $[\boldsymbol{l}, \boldsymbol{u}]$
4:     Sample $\boldsymbol{d}_0 \sim \mathcal{N}(0, I_{|\mathcal{C}|})$,    $\boldsymbol{c}_0 \sim \text{Uniform}(\{1, 2, \ldots, K\})^{|\mathcal{I}|}$
5:     $[\boldsymbol{d}_0, \boldsymbol{c}_0] \leftarrow \text{Project}_{\mathcal{B}}([\boldsymbol{d}_0, \boldsymbol{c}_0])$
6:     $\boldsymbol{G}_0 \leftarrow (\boldsymbol{G}, \boldsymbol{d}_0, \boldsymbol{c}_0)$
7:     **for** $dt \in [\Delta t_1, \Delta t_2, \cdots, \Delta t_{N_{\text{in}}}]$ **do**       ▷ Simulate Fokker-Planck & Kolmogorov Equations
8:         $p(\boldsymbol{d}_{1|t}), \boldsymbol{c}_{1|t} \leftarrow g_\theta(\boldsymbol{G}_0)$
9:                                                                          ▷ Integer Variable guidance
10:         Sample $\{\boldsymbol{d}_{1|t,1}, \boldsymbol{d}_{1|t,2}, \ldots, \boldsymbol{d}_{1|t,R}\} \sim p(\boldsymbol{d}_{1|t})$
11:         **for** $i = 1$ **to** $|\mathcal{I}|$ **do**
12:             $\hat{R}(d_t^{(i)}, \cdot) \leftarrow \dfrac{\sum_{r=1}^R f(\boldsymbol{d}_{1|t,r}, \boldsymbol{c}_{1|t}) R_{t|1}(d_t^{(i)}, \cdot | d_{1|t,r}^{(i)})}{\sum_{r=1}^R f(\boldsymbol{d}_{1|t,r}, \boldsymbol{c}_{1|t})}$
13:             Sample $d_{t+\Delta t}^{(i)} \sim \text{Cat}\left(\delta(d_{t+\Delta t}^{(i)}, d_t^{(i)}) + \hat{R}(d_t^{(i)}, d_{t+\Delta t}^{(i)}) \cdot \Delta t\right)$
14:         **end for**
15:                                                                          ▷ Continuous Variable guidance
16:         Sample $\boldsymbol{d}_{1|t} \sim \text{Cat}\left(f(\{\boldsymbol{d}_{1|t}^{(k)}\}_{k=1}^K) \cdot \boldsymbol{c}_{1|t}\right)$
17:         **for** $j = 1$ **to** $N_{\text{iter}}$ **do**
18:             $\boldsymbol{c}_t \leftarrow \text{Project}_{\mathcal{B}}\left(\boldsymbol{c}_t + \rho \nabla_{\boldsymbol{c}_t} f\left(\boldsymbol{d}_{1|t}, g_\theta(\boldsymbol{d}_t, \boldsymbol{c}_t)_{\boldsymbol{c}}\right)\right)$
19:         **end for**
20:         $\boldsymbol{c}_{1|t} \leftarrow g_\theta(\boldsymbol{d}_t, \boldsymbol{c}_t)_{\boldsymbol{c}}$
21:         $\hat{v}_t(\boldsymbol{c}_t) \leftarrow v_{t|1}(\boldsymbol{c}_t \mid \mathbb{E}_{1|t}[\boldsymbol{c}_{1|t} \mid \boldsymbol{G}_t]) = \frac{\boldsymbol{c}_{1|t} - \boldsymbol{c}_t}{1 - t}$
22:         $\boldsymbol{c}_{t+\Delta t} \leftarrow \text{Project}_{\mathcal{B}}(\boldsymbol{c}_t + \hat{v}_t(\boldsymbol{c}_t) \cdot \Delta t)$
23:         $t \leftarrow t + \Delta t$
24:     **end for**
25:     **Output:** $\boldsymbol{X}_1, \boldsymbol{a}_1$
26: **end procedure**

---

# G    ADDITIONAL EXPERIMENTS

## G.1    BREAKDOWN ANALYSIS ON MIPLIB 2017 BENCHMARK

To further investigate scalability and analyze failure cases, we partitioned the MIPLIB 2017 test set into five subsets based on the scale of the MILP instances. We report the relative primal gap, Rel.GAP, of FMIP on these subsets w.r.t. the Best Known Solution (BKS) provided by MIPLIB official website.

Table 8: Relative Primal Gap of FMIP on subsets of varying instance sizes.

| Instance Size | FMIP Rel. GAP | Best Baseline Rel. GAP |
|---|---|---|
| 50 – 3,000 | **4.70%** | 5.03% |
| 3,000 – 8,500 | **4.81%** | 5.19% |
| 8,500 – 20,000 | **4.86%** | 5.445% |
| 20,000 – 70,000 | **9.04%** | 9.10% |
| 70,000 – 300,000 | **23.26%** | 25.38% |

The results in Table 8 reveal a distinct performance dichotomy: while FMIP maintains robust stability (Rel. GAP $< 5\%$) across small to medium-large instances (up to 20,000 variables), performance degrades on ultra-large-scale instances ($> 70,000$ variables). In these extreme regimes, the GAP widens to $\sim 23\%$ as the exponentially increasing complexity of the constraint landscape challenges the model's generative precision.

Despite the degradation in extreme cases, FMIP still serves as the SOTA primal heuristic predictor to provide feasible starting points, making it significantly faster than a cold-start solver. This analysis provides a clear direction for future work, i.e., improving backbone scalability for ultra-large instances.

## G.2    SCALABILITY ON EXTREMELY HARD INSTANCES

We further test FMIP combined with Predict-and-Search (FMIP-PS) against Gurobi on extremely large-scale Set Covering (SC) problems (up to 1,000,000 variables), which are significantly larger and harder than the training set (around 5,000 variables). The time limit of FMIP is set to 3600s, which is empirically enough for superior performance in such extreme cases. The time limit of Gurobi is set according to the problem scale.

Table 9: Objective Value over Time w.r.t. Different Test Problem Scale (Lower is Better).

(a) #Constraints = 50,000; #Variables = 500,000

| Solver | 60s | 600s | 1800s | 3600s |
|---|---|---|---|---|
| Gurobi | 785,972 | 785,972 | 53,008 | 52,877 |
| FMIP-PS | 1,335,583 | **52,984** | **52,905** | **52,859** |

(b) #Constraints = 100,000; #Variables = 500,000

| Solver | 60s | 1800s | 3600s | 7200s |
|---|---|---|---|---|
| Gurobi | 1,015,955 | 1,015,955 | 1,015,955 | 1,015,955 |
| FMIP-PS | **1,007,549** | **1,007,549** | **76,579** | —Stop— |

(c) #Constraints = 500,000; #Variables = 500,000

| Solver | 60s | 1800s | 3600s | 7200s | 25000s |
|---|---|---|---|---|---|
| Gurobi | 1,654,768 | 818,547 | 469,130 | 469,130 | 469,130 |
| FMIP-PS | **1,007,549** | **184,821** | **184,821** | —Stop— | —Stop— |

(d) #Constraints = 100,000; #Variables = 1,000,000

| Solver | 60s | 1800s | 3600s | 7200s | 25000s |
|---|---|---|---|---|---|
| Gurobi | 1,022,627 | 1,022,627 | 1,022,627 | 1,022,627 | 44,039 |
| FMIP-PS | **192,417** | **192,417** | **43,113** | —Stop— | —Stop— |

We can observe that, for these hard instances, Gurobi often stagnates or converges very slowly. FMIP could help Gurobi to find solutions that are orders of magnitude better (e.g., $\sim 43\text{k}$ vs $\sim 1\text{M}$) or reaches comparable solutions hours faster. These results further demonstrate the superiority of our proposed FMIP to further boost Gurobi, and validate the core practice of accelerating downstream solvers with AI powers.

## H    RELATIONSHIPS BETWEEN FMIP AND EXACT SOLVERS LIKE GUROBI

We would like to further clarify that FMIP is designed to enhance exact downstream solvers like Gurobi, instead of replacing them.

The exact solver, Gurobi, is the backend engine in our experiments. The "Downstream Solvers" in our paper (i.e., Neural Diving, Predict-and-Search, Apollo) are, in fact, neural decomposition frameworks that wrap the exact solver Gurobi as improved version. They do not replace Gurobi; rather, they use the heuristic solution predicted by FMIP to decompose the original hard MILP into smaller sub-MIPs, which are then solved by Gurobi. Therefore, our method is designed precisely to function with wrapped or exact solvers, enhancing their ability to find high-quality feasible solutions in restricted search spaces.

