# OpenReview forum: "FMIP: Joint Continuous-Integer Flow For Mixed-Integer Linear Programming"
_ICLR.cc/2026/Conference — ICLR 2026 Poster_

### Official Review · Reviewer_rMCB · 2025-10-17

**Soundness:** 3
**Presentation:** 4
**Contribution:** 3
**Rating:** 6
**Confidence:** 4

**Summary:**

The authors study mixed-integer optimization problems (MILPS) which are computationally hard to solve to optimality in general. Solvers for MILPs benefit from good heuristic start solutions, which can be generated by using state-of-the-art learning methods. The authors propose a generative model which is able to generate good feasible solutions by predicting (compared to previous models) both, integer and continuous variables. The performance of the model is tested on a wide variety of instances and compared to several benchmarks.

**Strengths:**

The paper is very clearly written and all concepts are presented in a detailed but concise way. The methodology is very sophisticated and the idea of incorporating the continuous variables into the generative model is interesting and hence the whole paper provides an important contribution to the knowledge of the field. The experiments are extensive, although there are some subtle issues which should be improved.

**Weaknesses:**

1. The goal of the paper is to provide (exact) solvers with good feasible solutions. Hence I don't understand why there is no state-of-the-art exact MILP solver used as downstream solver (e.g. Gurobi, SCIP or HiGHS)? It would be important to see on very hard instances, how much the predicted solutions improve the performance of the exact solver (e.g. in terms of optimality gap after 1 hour). Solvers as Gurobi have sophisticated methods implemented to search for good feasible solutions. If your extremely complex model (which has to be trained etc.) cannot provide better solutions then it is useless for exact solvers.

2. In the experiments Gurobi is able to provide the same objective values as your model for nearly all instances (except LB and IP, although on LB it nearly achieves the same value). This indicates that the instances which are solved here are quite easy for exact solvers as Gurobi and at the same time Gurobi provides you with an optimality gap (which you should report). So looking at the results as a user I would prefer using Gurobi, which maybe has to run for 3600 seconds but on the other hand it is extremely simple to use: there is no training involved and it provides an optimality gap. As mentioned above, it would be interesting to see how good your predicted solutions are for harder instances.

3. It is not clear from the main paper, what data the model is trained on. Since you have to provide good solutions for the training instances these can only be instances which are easy to solve by the MILP solvers. Your model should be evaluated on instances which are of larger dimension/harder than the training set.

Minor comments:
- The paragraph "Metrics" is not clear enough. What exactly do you mean hear by OBJ and what is BKS? Is BKS the best known solution after running the downstream solver? Why do you compare it to OBJ which I understand is the objective value of the solution predicted by the generative models?
- The statement in abstract and conclusion that the primal gap is reduced by 41.34% is misleading. Gurobi is able to provide primal solutions nearly as good as your model in one hour.

**Questions:**

- Why don't you use any exact downstream solvers as Gurobi, SCIP etc.?
- Looking at the experiments, why should I use your (very complex) model involving data collection and training instead of just running Gurobi?
- How good is your model for hard instances which are harder than the instances in the training set?
- How much does your model improve the Gurobi process compared to the standard heuristics implemented in Gurobi?

---

> ### Author Response · Authors · 2025-11-23
> **Official Response to Reviewer rMCB - Part 1**
>
> We thank the reviewer for the positive comments on the clarity of our writing and the sophistication of our methodology. We appreciate the constructive challenge regarding the practical utility of our model compared to exact solvers, which motivated us to conduct additional large-scale experiments on "very hard" instances.
>
> We address the specific concerns and questions below.
>
> > ### **W1 & Q1**: Relationship with Exact Solvers like Gurobi
>
> FMIP is designed to enhance Gurobi, instead of replacing it.
>
> 1. **The exact solver, Gurobi, is the backend engine in our experiments**. The "Downstream Solvers" in our paper (i.e., Neural Diving, Predict-and-Search, Apollo) are, in fact, *neural decomposition* frameworks that **wrap the exact solver** Gurobi as improved version. They do not replace Gurobi; rather, they use the heuristic solution predicted by FMIP to decompose the original hard MILP into smaller sub-MIPs, which are then solved by Gurobi.
>     - **Workflow**: Input MILP Instance → FMIP Prediction → Neural Decomposition (defining sub-problems) → Gurobi (Exact Solver) → Final Solution.
>     - Therefore, our method is designed precisely to function with wrapped or exact solvers, enhancing their ability to find high-quality feasible solutions in restricted search spaces.
>
> 2. **Performance vs. Gurobi (3600s) on Hard Instances**. We have already reported the performance of FMIP against the exact solver. In Table 1, the row "Gurobi (3600s)" represents the standard Gurobi solver running for 1 hour. On some challenging benchmarks where Gurobi struggles to find good solutions even after 1 hour, our FMIP-guided frameworks largely outperform it with runtime of only 400~800 seconds. For examples, on Item Placement (IP) dataset, FMIP achieves a gap of 13.74 (using Apollo), significantly better than Gurobi's 17.73 despite using a fraction of the time (800s vs. 3600s).
>
> 3. **Lightweight Complexity & Seamless Integratation**. Finally, contrary to the concern about complexity, FMIP is designed as a lightweight, plug-and-play module that introduces negligible inference overhead (often milliseconds, see Table 3) while seamlessly empowering downstream wrapped or exact solvers to conquer hard instances.
>
> In the revised paper, we would update Section 5.1 to clarify that these "downstream solvers" rely on Gurobi as the exact solving engine, and emphasize the comparison against the standalone Gurobi baseline.
>
> > ### **W2**: Complexity and Practical Utility
>
> We thank the reviewer for the comments regarding usability and metrics. We respectfully disagree that the "easy-to-use simplicity" of Gurobi outweighs the significant computational advantages of our method, and we clarify our metric choice below.
>
> 1. **4x+ Acceleration & Superiority on Hard Instances**. FMIP reduces solving time by **~4.5x** (400~800s vs. 3600s) while matching Gurobi on standard tasks, and even significantly outperforming it on hard instances (e.g., Item Placement GAP: 13.74 vs. 17.73). As a lightweight, **plug-and-play** module with **negligible inference overhead** (usually milliseconds in Table 3), this substantial acceleration is critical for industrial applications, justifying the value of FMIP over waiting significantly longer for an exact solver.
>
> 2. **Regarding optimality gap**, we only report the Primal Gap (distance to Best Known Solution) instead of the Optimal Gap because FMIP acts as a primal heuristic. Its objective is to quickly identify high-quality feasible solutions (improving the upper bound), rather than tightening the dual bound/optimality gap. This is the standard metric for evaluating heuristic improvements [1, 2].
>
> > ### **W3**: Training Data Details
>
> We have already provided the details of training data in Appendix D.1. We follow standard protocols (9:1 split) on established benchmarks to ensure fair comparison with baselines. We acknowledge that this detail should be more prominent and will emphasize the exact training data settings in the main content of the paper.
>
> > ### **Minor Comments**: Metric Definition & Relative Improvement
>
> As stated in Line 318-323, Best Known Solution (BKS) is defined as the best objective found by any method in our own experiments (including one-hour Gurobi runs). OBJ is the objective value found by a MILP solving method, e.g., one heuristic predictor + one downstream solver (wrapped version of Gurobi).
>
> As for the 41.34% improvement, we apologize for the lack of explicit detail regarding this calculation of the overall improvement. As shown in Table 1, we report the relative improvement (Rel.Imprv.) of FMIP over the best baseline across 8 benchmarks and 4 downstream solvers. The overall improvement is computed by averaging all the 4*8=32 relative improvement results. We will clarify the details of such a 41.34% overall improvement in the revised paper.

---

> ### Author Response · Authors · 2025-11-23
> **Official Response to Reviewer rMCB - Part 2**
>
> > ### **Q2**: Why use AI instead of just Gurobi?
>
> We would like to again clarify that AI models, including our FMIP, are not intended to replace Gurobi, but to provide a better heuristic solution to **further accelerate** downstream wrapped/exact solvers like Gurobi. The rationale is that generic solvers (like Gurobi) use general-purpose heuristics. By training on specific data distributions, AI model captures structural patterns that generic heuristics miss, allowing it to solve hard instances that are intractable for the solver alone.
>
> This general idea has already been studied and deployed by top-tier research institutes and industry, such as Google [1], Meta [2], Huawei [3, 4], Alibaba [5], etc. Our proposed FMIP achieves the new SOTA performance by proposing the joint modeling framework on both integer and continuous variables.
>
> > ### **Q3**: Performance on Large-Scale Hard Instances
>
> We further test FMIP (combined with Predict-and-Search) against Gurobi on extremely large-scale Set Covering (SC) problems (up to **1,000,000** variables), which are significantly larger and harder than the training set (around **5,000** variables). The time limit of FMIP is set to 3600s, which is empirically enough for superior performance in such extreme cases.
>
> **Table**: Objective Value over Time w.r.t. Different Test Problem Scale (Lower is Better)
>
> (a) #Constraints = 50,000; #Variables = 500,000
>
> |  | 60s | 600s | 1800s | 3600s |
> |---|---|---|---|---|
> | Gurobi | **785972** | 785972 | 53008 | 52877 |
> | FMIP-PS | 1335583 | **52984** | **52905** | **52859** |
>
> (b) #Constraints = 100,000; #Variables = 500,000
>
> |  | 60s | 1800s | 3600s | 7200s |
> |---|---|---|---|---|
> | Gurobi | 1015955 | 1015955 | 1015955 | 1015955 |
> | FMIP-PS | **1007549** | **1007549** | **76579** | —Stop— |
>
> (c) #Constraints = 500,000; #Variables = 500,000
>
> ||60s|1800s|3600s|7200s|25000s|
> |-|-|-|-|-|-|
> |Gurobi|1654768|818547|469130|469130|469130|
> |FMIP-PS|**1007549**|**184821**|**184821**|—Stop—|—Stop—|
>
> (d) #Constraints = 100,000; #Variables = 1,000,000
>
> |  | 60s | 1800s | 3600s | 7200s | 25000s|
> |---|---|---|---|---|---|
> | Gurobi | 1022627 | 1022627 | 1022627 | 1022627 |44039|
> | FMIP-PS | **192417** | **192417** | **43113** | —Stop— | —Stop— |
>
> We can observe that, for these hard instances, Gurobi often stagnates or converges very slowly. FMIP could help Gurobi to find solutions that are **orders of magnitude better** (e.g., ~43k vs ~1M) or reaches comparable solutions hours faster. These results further demonstrate the superiority of our proposed FMIP to **further boost Gurobi**, and validate the core practice of accelerating downstream solvers with AI powers.
>
> > ### **Q4**: Improvement over Standard Gurobi Heuristics
>
> The overall performance in Table 1 in the original paper, as well as the additional experiments on extremely large-scale hard cases above, demonstrates that the predicted heuristic from FMIP could outperform the standard ones and further accelerate Gurobi.
>
> This is reasonable because FMIP functions in a data-driven way, incorporating more specific distributional information than the general-purpose heuristics coupled within the Gurobi solving phase.
>
> ### **Reference**
>
> [1] Nair, V., et al. "Solving Mixed Integer Programs Using Neural Networks." arXiv 2020.
>
> [2] Huang, T., et al. "Contrastive Predict-and-Search for Mixed Integer Linear Programs." ICML 2024.
>
> [3] Han, Q., et al. "A GNN-Guided Predict-and-Search Framework for Mixed-Integer Linear Programming." ICLR 2023
>
> [4] Liu, H., et al. "Apollo-MILP: AnAlternating Prediction-Correction Neural Solving Framework for Mixed-Integer Linear Programming." ICLR 2025
>
> [5] Zhang, M., et al. "MindOpt Tuner: Boost the Performance of Numerical Software by Automatic Parameter Tuning." arXiv 2023.

---

### Official Review · Reviewer_t9Xo · 2025-10-29

**Soundness:** 3
**Presentation:** 3
**Contribution:** 2
**Rating:** 4
**Confidence:** 4

**Summary:**

FMIP uses a multimodal flow matching generative model to learn the distribution of high-quality MILP solutions in a combined continuous–discrete space. The author proposes a **“holistic” guidance mechanism** that steers the generation process using the MILP’s objective function and constraint satisfaction feedback, refining candidate solutions toward feasibility and optimality during sampling. This approach is expected to addresses the limitation of previous graph neural network (GNN) predictors that predicted only integer variables and left continuous ones to a solver, thereby missing the intricate coupling between them .

**Strengths:**

1. The joint modeling of integer variables and continuous variables is a notable contribution, where previous works usually only focus on discrete value predictions, which ignores some global relationships.
2. The author proposed the holistic guidance mechanism integrated into FMIP’s sampling. During inference, FMIP uses the MILP’s objective and constraint violations to guide the generative trajectory, which is expected to produce better predictions.

**Weaknesses:**

1. Given the context where integer–continuous coupling is important, the motivation for a generative approach is under-developed. Why is a generative model preferable to discriminative predictors under identical solver budgets? Why is the Flow Matching used in the proposed framework?
2. In Tables 1–2 the downstream solver time limits should be standardized (e.g., a flat 1000 s for all methods) to ensure fairness; the mixed 400/600 s budgets are short for some instances and confound comparisons. A unified budget would make the FMIP gains more persuasive.

**Questions:**

1. The guidance combines the objective with squared constraint violations via a hyperparameter $\gamma$. Please clarify how it balances objective vs. violation terms, its sensitivity, and the tuning/selection procedure used.
2. The reweighting step appears to require evaluating continuous variables, which is extra computational efforts. If many continuous variables are functionally dependent or cheaply recoverable, why is explicit prediction necessary? Could a GNN/GAT that aggregates global information suffice without predicting continuous components? Is it necessary because of the generative model? Please clarify the additional computational cost or justify the benefit of jointly predicting discrete and continuous variables (e.g., tighter guidance, reduced repair time, better warm starts).

---

> ### Author Response · Authors · 2025-11-23
> **Official Response to Reviewer t9Xo - Part 1**
>
> We thank the reviewer for the thoughtful assessment, as well as the recognition of the value of our joint modeling contribution and the holistic guidance mechanism. We would like to address your concerns as follows.
>
> > ### **W1**: Motivation for Generative Models & Flow Matching
>
> We appreciate the opportunity to clarify our motivation for generative flow matching approach. Our rationale is two-fold:
> 1. **Why generative instead of discriminative?** A fundamental characteristic of MILP is that a single instance often admits multiple global optima or a diverse set of high-quality local optima (i.e., an intrinsic "**one-to-many**" nature). Consequently, heuristic prediction is best formulated as a **distribution modeling problem** rather than a single-point prediction task. Discriminative models, which tend to converge to a deterministic "average" of the training data, often fail to capture this complexity of underlying distribution, resulting in solutions that lie in low-density or even infeasible regions.
> 2. **Within generative models, why flow matching?** We select Flow Matching due to its simulation-free training and stable, straight-line vector field objectives, which offer a simpler and more direct alternative to diffusion-based methods. However, we emphasize that our core contributions (i.e., **Joint Modeling Principle** and **Holistic Guidance Mechanism**) are **backbone-agnostic**. Flow Matching serves as an efficient instantiation of our framework, but the principles & guidance mechanism can be readily generalized to other generative schemes like diffusion.
>
> Moreover, regarding the identical budget concern, we note that the inference overhead of our generative model is negligible compared to the solver's runtime (less than 4.8 seconds vs. hundreds/thousands of seconds) in Section 5.4. The marginal increase in inference time yields a substantial reduction in total solving time, validating our choice for the generative flow matching approach.
>
> > ### **W2**: Evaluation protocol regarding the time budgets for different solvers
>
> We appreciate the suggestion regarding standardization. However, we would like to clarify our experimental design and the rationale behind the chosen time limits:
> 1. **Internal Consistency & Fairness**: Our framework is **decoupled** from the downstream solver. The core objective is not to compare different solvers against each other, but to evaluate the acceleration capability of FMIP *given a specific solver context*. Crucially, within each experimental setting (i.e., for a specific dataset and a downstream solver), the time limit is kept **strictly identical** for all compared methods (FMIP and other baselines SL, IP-Guided-Diff, DIFUSCO). Therefore, the comparison remains fair and valid within each group.
> 2. **Standard Practice & Solver Heterogeneity**: Setting different time budgets for different downstream solvers (e.g., 400s vs. 600s vs. 800s) is a **common practice** in neural combinatorial optimization literature [1, 2]. This is necessary because different neural solver methods have vastly different performance scales. A "one-size-fits-all" budget (e.g., 1000s) might be too loose for a strong solver or too tight for a weaker one, obscuring the true impact of the warm-start heuristic. Our settings are chosen to align with established benchmarks to ensure meaningful evaluation.

---

> ### Author Response · Authors · 2025-11-23
> **Official Response to Reviewer t9Xo - Part 2**
>
> > ### **Q1**: Impact of Guidance Coefficient $\gamma$
>
> In our reported experiments, we used a fixed coefficient $\gamma$=50 in Eq. 5 to balance the objective- and feasibility-based guidance signals, which is a hyperparameter selected from {0.1, 1, 10, 50, 100}.
>
> As also suggested by Reviewer n8PZ, a fixed coefficient $\gamma$ might lead to an imbalance where one term dominates the guidance during the iterative generation process. To investigate this, we conduct additional experiments by introducing an **adaptive balancing strategy** for the coefficient $\gamma$ to replace the previously fixed strategy. Specifically, at each generation step, we compare the magnitude of the objective term with the constraint-violation term:
> - If the objective term is larger, we scale up $\gamma$ by a factor $\alpha$ (i.e., $\gamma \leftarrow \gamma \times \alpha$) to enforce stronger constraint satisfaction.
> - Conversely, if the constraint term is large, we scale down $\gamma$ by a factor $\alpha$ (i.e., $\gamma \leftarrow \gamma / \alpha$).
>
> This adjustment is cumulative across the sampling steps (e.g., $\gamma \times \alpha \times \alpha$…), allowing the model to dynamically recalibrate the guidance focus between optimality and feasibility. We evaluate this adaptive strategy on the Item Placement (IP) dataset using PS as the downstream solver, varying both the scaling factor $\alpha$ and the number of sampling steps.
>
> **Table**: Impact of Adaptive Balancing Strategy on Objective Value (Lower is Better)
> |**Strategy**|**Step-12**|**Step-24**|**Step-32**|
> |-|-|-|-|
> |Fixed $\gamma$|**11.79**|12.39|13.25|
> |$\alpha=1.1$|11.93|12.35|12.05|
> |$\alpha=1.5$|11.99|**11.85**|12.68|
> |$\alpha=2.0$|12.19|11.95|**11.92**|
>
> **Result Analysis**. With a fixed $\gamma$, the objective value degrades noticeably as sampling steps increase (from 11.79 at Step-12 to 13.25 at Step-32). However, adaptive balancing strategy ($\alpha>1$) effectively mitigates this performance degradation at larger steps. For instance, with $\alpha=2.0$, the performance at Step-32 (11.92) is significantly better than the fixed version (13.25) and remains competitive with lower step counts.
>
> **Conclusion & Future Work**. These results validate the reviewer's hypothesis: *dynamically balancing the guidance terms is crucial for stabilizing generation over longer trajectories*. While the current optimum in this preliminary experiment is still achieved by the fixed $\gamma$ at Step-12 (11.79), the trend suggests that the dynamic strategy prevents the guidance imbalance issue. We believe that with a more sophisticated adaptive strategy (beyond simple scaling), we can unlock the full potential of longer sampling steps to surpass current best results. We identify this as a critical and promising direction for our future work.
>
> We will include the experiment and discussion in the revised paper.
>
> > ### **Q2**: Necessity & Efficiency of Explicit Continuous Prediction
>
> We thank the reviewer for questioning the necessity and computational trade-offs of explicitly predicting continuous variables. We clarify that this design choice is essential for our guidance mechanism and is computationally far more efficient than the alternative of "recovery".
>
> 1. **Necessity for Holistic Guidance & Performance**. Explicitly predicting continuous variables is critical for our holistic guidance mechanism during the generation process, which leads to more accurate guidance, better warm starts, and thereby faster MILP solving. The overall performance in Table 1 and the "w/o Continuous" variant in Table 4 both confirm that capturing the joint distribution via explicit continuous prediction is essential for high-quality solution prediction.
>
> 2. **Efficiency: Prediction vs. Recovery**. The reviewer suggests "recovering" continuous variables might be cheaper. We argue that explicit prediction is significantly faster in a generative context.
>     - **Matrix Multiplication vs. LP Solving**: "Recovering" continuous variables typically requires solving a Linear Programming (LP) relaxation. Doing this iteratively at every step of the flow matching process would be computationally prohibitive. In contrast, FMIP’s explicit prediction relies solely on efficient matrix multiplications within the neural network.
>     - **Negligible Inference Overhead**: As shown in Table 3, the entire model inference time of FMIP is comparable to integer-only baselines (e.g., DIFUSCO), and even negligible when considering the runtime of downstream solvers (<5s vs. often hundreds/thousands of seconds). Explicit prediction thus nearly provides a "free lunch": it enables tighter holistic guidance and better heuristic solution without introducing massive additional computational cost.
>
> ### **Reference**
>
> [1] Zhang, C., et al. "Deep Reinforcement Learning Guided Improvement Heuristic for Job Shop Scheduling." ICLR 2024.
>
> [2] Sun, Z. and Yang, Y. "DIFUSCO: graph-based diffusion solvers for combinatorial optimization." NeurIPS 2023.

---

> ### Comment · Reviewer_t9Xo · 2025-11-26
>
> I appreciate the clarification from the authors. All of my concerns have been addressed, and I would like to raise my score.

---

> > ### Author Response · Authors · 2025-11-26
> > **Thanks for Your Feedback!**
> >
> > We sincerely appreciate your feedback and are glad to hear that our response has effectively addressed your concerns. We are encouraged by your decision to raise the score.
> >
> > Your constructive suggestions, particularly regarding the adaptive balancing strategy for the guidance coefficient, have significantly helped us strengthen the robustness of our framework. We are committed to incorporating these additional experiments and clarifications into the final version of the paper.
> >
> > Thank you again for your time and support!

---

### Official Review · Reviewer_n8PZ · 2025-10-31

**Soundness:** 3
**Presentation:** 3
**Contribution:** 2
**Rating:** 4
**Confidence:** 3

**Summary:**

This paper introduces FMIP a flow model for MILP solving, cast at sampling from a generative model, that defines a method to sample from joint distributions over discrete and continuous variable assignments. This can help capturing interactions between these two sets of variables, which is often missing in previous works on generative models for MILPs.
The model is based on a 'tripartite-graph'  a variation of the standard bipartite graph where integer and continuous variables are mapped to two disjoint set of nodes.

However, while the distribution is joint, it is assumed to be factorized and amounts to one independent prediction per variable (discrete or continuous). Only the conditioning is from a complete, and noisy, assignment.

Additionally, authors introduce guidance mechanisms for the two types of
variables helping steer the model's trajectories towards feasible assignments.
For the continuous variables the guidance is based on projected gradient descent to enforce box constraints (the other constraints are taken into account via the violation penalty)
For integer variables, authors use a sample-and-reweight method where the denoiser is used to generate several candidates which are evaluated. The rates are updated to promote (ie, increase the probability) of candidates which combine good objective value and low constraint violation. The final samples are also projected to enforce box constraints.



Eventually, FMIP returns an assignment for the continuous relaxation of the
input MILP, and must be processed further by a solver. In that sense, FMIP
remains a tool to warm-start a solver.


Typos:
- The two first references to Gasse et al. are the same
- The two first references to Taoan Huang et al. are the same
- l. 712 missing space before Edges
- l. 728 why $> 0$ and not $\neq 0$ ?
- l. 753 $t$ and $t$ ??

**Strengths:**

This paper identifies and  highlights one of the blind posts from previous generative models for MILPs: the lack of a proper treatment  of continuous variables. I believe this is the main contribution of the paper

**Weaknesses:**

- The model is straightforward and amounts to the concatenation of a continuous flow and a discrete flows that can both be conditioned on predicted assignments.
- The tri-partite representation is surprising and questionable (see question below)

**Questions:**

1. Where is the 41.34% improvement mentioned in the abstract and the conclusion found in the experiments? Maybe it's lost
2. In Table 4, how is the system "w/o Continuous" implemented? How does it handle continuous variables? This is important since, as mentioned by authors, this performance gap gives a direct evidence that modelling joint distributions as advocated in FMIP is important.
3. Why do you call your graph a tripartite graph instead of a bipartite graph with continuous/discrete binary features for variable nodes? It overcomplicates the model and makes the reader thinks edges between discrete and continuous nodes are allowed.
   Besides, line 726 mentions a bipartite graph
4.The discussion lines 468-474 is interesting. Could an additional reason to explain why the number of sampling steps may have a detrimental effect is that the coefficient $\gamma$ given to the violation (and possibly the measure of the violation itself based on the max operator) may lead to a situation where the function $f$ favors constraints over objective?

---

> ### Author Response · Authors · 2025-11-23
> **Official Response to Reviewer n8PZ - Part 1**
>
> We sincerely thank the reviewer for the detailed analysis and constructive feedback. We would like to address the concerns as follows.
>
> > ### Typos
>
> We sincerely thank the reviewer for pointing out the typos. We have corrected the duplicate references and the typos in the revised version.
>
> > ### **W1**: FMIP is straightforward and amounts to the concatenation of continuous & discrete flows.
>
> We would like to clarify that the primary contribution of our work **extends beyond** the architectural design of a mixed flow network (i.e., simply concatenating flows). Instead, our core contribution lies in identifying and validating a crucial, previously overlooked **design principle** for MILP heuristic prediction: **joint modeling of both continuous and integer variables**. The mixed flow architecture serves as the necessary implementation vehicle for this principle, which in turn unlocks technically significant capabilities such as the holistic guidance mechanism.
>
> 1. **The Value of Joint Modeling**: To the best of our knowledge, we are **the first** to explicitly propose the joint modeling of integer and continuous variables for MILP heuristic prediction. The fact that this "Core Principle" has been overlooked in existing literature—yet proves to be highly effective in our experiments—highlights a critical gap. Identifying and validating this missing link is a primary contribution of our work.
> 2. **Non-trivial Technical Realization**: Beyond the conceptual level, the technical challenges and solutions stemming from this principle are non-trivial and offer significant depth:
>     - **Generative Distribution Modeling**: By introducing Flow Matching, we shift the paradigm from single solution prediction to modeling & generating the full distribution of heuristics. This aligns better with the inherent "one-to-many" challenge of MILP, where a single instance may correspond to multiple high-quality solutions.
>     - **Holistic Guidance Mechanism**: The joint modeling approach facilitates a holistic guidance mechanism for both feasibility- and objective-based guidance. This remains an unsolved challenge for previous works, since partial modeling fails to capture the full variable space, rendering such holistic guidance impossible.
>
> Following the valuable feedback, we will revise the Introduction, Related Works, and Methodology sections to better emphasize the novelty and contribution of our FMIP framework.
>
> > ### **W2 & Q3**: Tripartite vs. Bipartite Graph Terminology
>
> We agree with the reviewer that, strictly topologically, the graph is bipartite (Variable Nodes $\leftrightarrow$ Constraint Nodes). There are no direct edges between integer and continuous nodes. We use this term "tripartite" to emphasize that integer and continuous nodes are treated as distinct classes with separate neural parameters and distinct flow dynamics (Eq. 3 vs. Eq. 6).
>
> To avoid confusion, we will remove the term "tripartite" and clarify in the paper that the graph topology is bipartite, but the parameterization distinguishes three node types to handle their heterogeneous nature. Thanks for the constructive feedback on this important terminology.
>
> > ### **Q1**: Calculation of the 41.34% Overall Improvement
>
> We apologize for the lack of explicit detail regarding this calculation of the overall improvement. As shown in Table 1, we reported the relative improvement (Rel.Imprv.) of FMIP over the best baseline across 8 benchmarks and 4 downstream solvers. The overall improvement is computed by averaging all the 4*8=32 relative improvement results. We will clarify the details of such a 41.34% overall improvement in the revised paper.
>
> > ### **Q2**: "w/o Continuous" Ablation Details
>
> The question is truly insightful! The "w/o Continuous" variant is restricted to generating only integer variables by blocking the continuous generation head and the continuous flow updates. Hence, it **does not and cannot** perform any guidance during the iterative generation process, since holistic guidance mechanism in Eq. 5 strictly requires both integer and continuous variables.
>
> The severe performance degradation of this "w/o Continuous" variant (the worst-performing in Table 4) provides the most direct evidence for our central claim: **joint modeling is the essential prerequisite for effective, holistic guidance**, and this guidance is critical for generating high-quality solutions for MILP problems.
>
> We will revise Section 5.5 to clarify the detailed implementation of "w/o Continuous" variant regarding the disabled guidance mechanism.

---

> > ### Author Response · Authors · 2025-11-23
> > **Official Response to Reviewer n8PZ - Part 2**
> >
> > > ### **Q4**: Sampling Steps & Impact of Coefficient $\gamma$ for Guidance Balancing
> >
> > Thanks for your profound insights, which inspire our extended experiments and potential future works!
> >
> > We agree that a fixed coefficient $\gamma$ in Eq. 5 might lead to an imbalance where one term dominates the guidance during the iterative generation process. To investigate this, we conduct additional experiments by introducing an **adaptive balancing strategy** for the coefficient $\gamma$ to replace the previously fixed strategy ($\gamma$=50). Specifically, at each generation step, we compare the magnitude of the objective term with the constraint-violation term:
> > - If the objective term is larger, we scale up $\gamma$ by a factor $\alpha$ (i.e., $\gamma \leftarrow \gamma \times \alpha$) to enforce stronger constraint satisfaction.
> > - Conversely, if the constraint term is large, we scale down $\gamma$ by a factor $\alpha$ (i.e., $\gamma \leftarrow \gamma / \alpha$).
> >
> > This adjustment is cumulative across the sampling steps (e.g., $\gamma \times \alpha \times \alpha$…), allowing the model to dynamically recalibrate the guidance focus between optimality and feasibility. We evaluate this adaptive strategy on the **Item Placement** (IP) dataset using **PS** as the downstream solver, varying both the scaling factor $\alpha$ and the number of sampling steps.
> >
> > **Table**: Impact of Adaptive Balancing Strategy on Objective Value (Lower is Better)
> > |**Strategy**|**Step-12**|**Step-24**|**Step-32**|
> > |-|-|-|-|
> > |Fixed $\gamma$|**11.79**|12.39|13.25|
> > |$\alpha=1.1$|11.93|12.35|12.05|
> > |$\alpha=1.5$|11.99|**11.85**|12.68|
> > |$\alpha=2.0$|12.19|11.95|**11.92**|
> >
> > **Experimental Results and Analysis**. With a fixed $\gamma$, the objective value degrades noticeably as sampling steps increase (from 11.79 at Step-12 to 13.25 at Step-32). However, adaptive balancing strategy ($\alpha>1$) effectively mitigates this performance degradation at larger steps. For instance, with $\alpha=2.0$, the performance at Step-32 (11.92) is significantly better than the fixed version (13.25) and remains competitive with lower step counts.
> >
> > **Conclusion & Future Work**. These results validate the reviewer's hypothesis: *dynamically balancing the guidance terms is crucial for stabilizing generation over longer trajectories*. While the current optimum in this preliminary experiment is still achieved by the fixed $\gamma$ at Step-12 (11.79), the trend suggests that the dynamic strategy prevents the guidance imbalance issue. We believe that with a more sophisticated adaptive strategy (beyond simple scaling), we can unlock the full potential of longer sampling steps to surpass current best results. We identify this as a critical and promising direction for our future work.
> >
> > We will include the experiment and discussion in the revised paper.
> >
> > Sincere thanks again for your insightful suggestions and inspirations! We are keen for further discussion and would appreciate any further recommendations on how to strengthen our manuscript.

---

> ### Comment · Reviewer_n8PZ · 2025-11-26
>
> Thank you to the authors for their detailed response.
> While I am still not convinced by the argument about "Non-trivial Technical Realization", I agree with the claim that the core contribution is the value of the joint modelling.
>
> I will keep my score as is.

---

> > ### Author Response · Authors · 2025-12-04
> > **Respectful Yet Different View on Technical Contribution: Simplicity as a Feature, Not a Weakness**
> >
> > We sincerely thank the reviewer for the continued engagement and for **acknowledging the value of our core contribution regarding joint modeling**.
> >
> > Regarding the "Technical Realization", we respect your perspective, but we hold a respectful yet different view on *what constitutes a high-quality technical contribution*. We believe that "**simple yet effective**" is often the gold standard for research.
> >
> > Much like the design philosophy behind foundational works (e.g., ResNet's skip connections), the technical value of a proposed method need not stem from a complex architecture or an intricate training pipeline. Instead, we argue that the simplicity of FMIP's technical design (the *holistic guidance* over mixed flow) is a **feature, not a weakness**:
> >
> > 1. **Efficiency & Robustness**: The "straightforward" implementation of the mixed flow validates that our identified core principle (joint modeling) is robust enough to work without needing elaborate architectural bells and whistles.
> > 2. **Generalizability**: A streamlined design is far easier for the community to reproduce, adopt, and extend to other generative schemes (e.g., diffusion models) or solver components.
> >
> > Therefore, we believe the fact that a mathematically clean and architecturally simple design can yield such significant improvements (41.34%) serves as strong evidence of FMIP's technical soundness and practical value.
> >
> > We hope this perspective helps clarify why we consider FMIP's technical design to be a significant strength, rather than a weakness. Thanks again for your time and valuable review.

---

### Official Review · Reviewer_1do2 · 2025-10-31

**Soundness:** 4
**Presentation:** 4
**Contribution:** 2
**Rating:** 6
**Confidence:** 3

**Summary:**

This manuscript is an incremental extension of [1], with the following core enhancements: i) an extention from integer linear programs (ILP) to mixed-integer linear programs (MILP), and ii) replacing diffusion models with flow matching models. The authors conduct extensive experiments, demonstrating improvements over existing methods.


[1] Zeng, Hao, et al. "Effective generation of feasible solutions for integer programming via guided diffusion." ACM SIGKDD 2024.

**Strengths:**

1. The paper is well-organized with a clear logical flow.
2. Extending the application of generative methods from ILP to MILP is a natural progression.
3. Empirical experiments are comprehensive, and the results show improvements over the selected baselines.

**Weaknesses:**

I have reviewed this paper at NeurIPS 2025, and I am satisfied with the revisions made by the authors. I do not have any major concerns, but I have a minor suggestion:

The paper states that "existing generative methods for MILP suffer from a critical limitation: they model the distribution of only the integer variables." However, the transition from ILP to MILP seems relatively straightforward, and this limitation may not constitute a major challenge. I recommend that the authors emphasize the more difficult aspects and make the challenges clearer to the reader.

**Questions:**

1. Does the inference time in Table 3 include the time cost of the guidance process?
2. In Table 4, how does the "w/o Continuous" variant perform the guidance? Without continuous variables, how are constraint violations and objective value calculated?

---

> ### Author Response · Authors · 2025-11-23
> **Official Response to Reviewer 1do2**
>
> Thanks for the reviewer's constructive feedback and we are encouraged that the reviewer found the paper well-organized, the presentation clear, and the experiments comprehensive. We would like to address the concerns as follows.
>
> > ### **Weakness**: transition from ILP to MILP is straightforward, and this limitation may not constitute a major challenge.
>
> We acknowledge the reviewer's perspective that the extension from ILP to MILP may appear conceptually straightforward. However, we respectfully argue that **this simplicity is exactly what makes our contribution both foundational and effective**.
>
> 1. **The Value of Joint Modeling**: While the transition seems intuitive, to the best of our knowledge, we are **the first** to explicitly propose the joint modeling of integer and continuous variables for MILP heuristic prediction. The fact that this "Core Principle" has been overlooked in existing literature—yet proves to be highly effective in our experiments—highlights a critical gap. Identifying and validating this missing link is a primary contribution of our work.
> 2. **Non-trivial Technical Realization**: Beyond the conceptual level, the technical challenges and solutions stemming from this principle are non-trivial and offer significant depth:
>     - **Generative Distribution Modeling**: By introducing Flow Matching, we shift the paradigm from single solution prediction to modeling & generating the full distribution of heuristics. This aligns better with the inherent "one-to-many" challenge of MILP, where a single instance may correspond to multiple high-quality solutions.
>     - **Holistic Guidance Mechanism**: The joint modeling approach facilitates a holistic guidance mechanism for both feasibility- and objective-based guidance. This remains an unsolved challenge for previous works, since partial modeling fails to capture the full variable space, rendering such holistic guidance impossible.
>
> Following the valuable feedback, we will revise the Introduction, Related Works, and Methodology sections to better emphasize the core challenges, as well as the novelty and contribution of our joint modeling framework.
>
> > ### **Q1**: Does the inference time in Table 3 include the time cost of the guidance process?
>
> Yes. The inference time reported in Table 3 absolutely includes the time cost of the full guidance process, showing the entire cost of heuristic prediction.
>
> The total time required to solve a MILP instance comprises 1) the inference time to predict heuristic solutions and 2) the runtime of the downstream solver. Although FMIP entails higher inference latency compared to baselines, this difference **is negligible** as the inference time (0.3s–4.8s) is **orders of magnitude smaller** than the solver's runtime (hundreds to thousands of seconds). Given that FMIP could achieve SOTA performance, we believe this slight increase in inference time is a highly favorable trade-off for practical usage.
>
> > ### **Q2**: "w/o Continuous" Ablation Details
>
> The "w/o Continuous" variant **does not and cannot** perform any guidance during the iterative generation process.
>
> Our holistic guidance mechanism in Eq. 5 is fundamentally dependent on evaluating the full MILP objective and constraints using a complete solution $(\boldsymbol{d},\boldsymbol{c})$. The "w/o Continuous" variant is an integer-only flow model that does not generate the continuous variables $\boldsymbol{c}$. Therefore, the guidance function $f(\boldsymbol{d},\boldsymbol{c})$ cannot be computed, and the entire guidance mechanism is necessarily disabled.
>
> The severe performance degradation of this "w/o Continuous" variant (the worst-performing in Table 4) provides the most direct evidence for our central claim: **joint modeling is the essential prerequisite for effective, holistic guidance**, and this guidance is critical for generating high-quality solutions for MILP problems.
>
> We will revise Section 5.5 to clarify the detailed implementation of "w/o Continuous" variant regarding the disabled guidance mechanism.

---

> > ### Comment · Reviewer_1do2 · 2025-11-24
> >
> > Thank you to the authors for their rebuttal. All of my concerns have been addressed, and I will maintain my vote for acceptance.

---

> > > ### Author Response · Authors · 2025-11-26
> > > **Thanks for Your Feedback!**
> > >
> > > Thank you very much for your follow-up and for supporting acceptance of our paper. We truly appreciate the time and effort you have spent reviewing our work and engaging with our rebuttal.
> > >
> > > Since the concern regarding the incremental nature of our work has been addressed, we kindly ask if you would consider re-evaluating the ```Contribution``` score (currently scored as 2) to reflect the non-trivial challenges and technical depth of our proposed FMIP framework. We would be grateful if you could kindly reconsider the overall evaluation.
> > >
> > > Thanks again for your time and effort!

---

### Official Review · Reviewer_3TSm · 2025-11-01

**Soundness:** 3
**Presentation:** 3
**Contribution:** 3
**Rating:** 6
**Confidence:** 2

**Summary:**

The paper introduces FMIP (Joint Continuous-Integer Flow for Mixed-Integer Linear Programming), a generative framework that addresses limitations in existing MILP heuristics by modeling the joint distribution of both integer and continuous variables using conditional flow matching. This approach captures the interdependencies between variable types, enabling a holistic guidance mechanism during inference that steers solutions toward optimality and feasibility via objective function and constraint feedback. FMIP, integrated with various graph neural network backbones and downstream solvers, is evaluated on some benchmarks.

**Strengths:**

The claimed empirical performance is good, achieving decent relative improvement in primal gap over state-of-the-art baselines on some benchmarks. The framework's compatibility with multiple GNN architectures and solvers, as shown in ablation studies and compatibility analyses, highlights its flexibility and generalizability. Additionally, the holistic guidance, supported by gradient-based updates for continuous variables and sampling-reweighting for integers, effectively leverages complete solution candidates at each step, leading to higher-quality heuristics.

**Weaknesses:**

FMIP's focus on bounded integer variables limits its applicability to general MILP problems with unbounded or non-binary integers. The reliance on flow matching introduces higher inference times compared to discriminative baselines like supervised learning. While ablations confirm the value of joint modeling and guidance, the paper could benefit from deeper analysis on failure cases or scalability to very large-scale instances beyond the tested benchmarks.

**Questions:**

None

---

> ### Author Response · Authors · 2025-11-23
> **Official Response to Reviewer 3TSm**
>
> We sincerely thank the reviewer for the positive assessment of our work, particularly for recognizing the framework's flexibility, the effectiveness of the holistic guidance mechanism, and our solid empirical results. We would like to address the reviewer's concerns as follows.
>
> > ### **W1**: FMIP's focus on  bounded integer variables limits its applicability to general MILP problems with unbounded or non-binary integers.
>
> We would like to clarify that FMIP is **not limited** to bounded binary variables.
>
> 1. FMIP is applicable to **non-binary** variables. As defined in Section 3.1 and Equation 2, FMIP explicitly handles general discrete variables with cardinality $K$, where $x_I \in \{0, 1, ..., K\}$, and predicts a categorical distribution over these $K$ states. This makes FMIP natively applicable to general integer variables, not just binary ones.
>
> 2. FMIP is applicable to **unbounded** integer variables. Previous works [1, 2] have successfully addressed unbounded integer variable modeling via engineering techniques like bit-wise prediction or continuous relaxation, which are fully compatible with our FMIP framework. In this paper, the involved benchmarks generally focus on bounded integers, since integer variables in real-world applications are naturally bounded by physical or logical constraints (e.g., time windows, vehicle capacities).
>
> > ### **W2**: Inference Time of Flow Matching
>
> The total time required to solve a MILP instance comprises 1) the model inference time to predict heuristic solutions and 2) the runtime of the downstream solver. Although flow matching entails higher inference latency compared to supervised learning, this difference **is negligible** as the inference time is **orders of magnitude smaller** than the solver's runtime.
>
> As detailed in Table 3 in the paper (copied below for your convenience), FMIP's inference time (0.3s–4.8s) is negligible compared to the downstream solver's runtime (hundreds to thousands of seconds). Furthermore, given that FMIP significantly outperforms SL by narrowing the primal gap (e.g., reducing it from 13.77% to 4.15% on MIPLIB with Apollo), we believe this slight increase in inference time is a highly favorable trade-off for practical usage.
>
> Table 3: Inference time (s) per MILP instance.
>
> |Downstream Solver|Training Method|CA|GIS|MIS|FCMNF|SC|LB|IP|MIPLIB|
> |-|-|-|-|-|-|-|-|-|-|
> |ND, PS, PMVB      |SL|0.047|0.089|0.138|0.065|0.087|0.088|0.023|0.115|
> ||DIFUSCO|0.187|0.652|0.340|0.238|0.412|1.154|0.093|1.561|
> ||IP-Guided-Diff|0.213|0.712|0.452|0.374|0.533|1.781|0.117|2.153|
> ||FMIP|0.281|0.761|0.476|0.310|0.623|1.294|0.176|2.043|
> |Apollo      |SL|0.082|0.273|0.184|0.310|0.209|0.213|0.093|0.327|
> ||DIFUSCO|0.512|2.125|1.341|0.545|1.112|3.129|0.153|3.231|
> ||IP-Guided-Diff|0.612|2.173|1.549|0.634|1.268|3.151|0.167|4.185|
> ||FMIP|0.718|2.060|1.805|0.546|1.782|3.229|0.121|4.812|
>
> > ### **W3**: Scalability Analysis & Failure Modes
>
> To further investigate scalability and analyze failure cases, we partitioned the **MIPLIB 2017** test set into five subsets based on the scale of the MILP instances. We report the *relative primal gap* of FMIP on these subsets w.r.t. the Best Known Solution (BKS) provided by MIPLIB official website.
>
> |**Subset ID**|**Instance Size**|**FMIP Rel.GAP**|**Best Baseline Rel.GAP**|
> |-|-|-|-|
> |1|50 - 3,000|4.70%|5.03%|
> |2|3,000 - 8,500|4.81%|5.19%|
> |3|8,500 - 20,000|4.86%|5.445%|
> |4|20,000 - 70,000|9.04%|9.10%|
> |5|70,000 - 300,000|23.26%|25.38%|
>
> The results are reported in the table above, from which we can observe:
> - **Robustness:** FMIP maintains stable and high performance (Rel.GAP < 5%) for small to medium-large instances (up to 20,000 variables).
> - **Failure Mode:** We observe performance degradation on very large-scale instances (>70,000 variables), where the Rel.GAP increases to ~23%. As the problem scale increases, the constraint landscape becomes exponentially more complex, requiring higher generative precision.
>
> Despite the degradation in extreme cases, FMIP still serves as the SOTA primal heuristic predictor to provide feasible starting points, making it significantly faster than a cold-start solver. This analysis provides a clear direction for future work, i.e., improving backbone scalability for ultra-large instances.
>
> ### **Reference**
>
> [1] Nair, V., et al. "Solving Mixed Integer Programs Using Neural Networks." arXiv 2020.
>
> [2] Huang, T., et al. "Contrastive Predict-and-Search for Mixed Integer Linear Programs." ICML 2024.
>
> [3] Gleixner, A., et al. "MIPLIB 2017: data-driven compilation of the 6th mixed-integer programming library." MPC 2021.

---

### Author Response · Authors · 2025-12-04
**Summary of Rebuttal**

We sincerely thank the reviewers for their constructive feedback. We would like to first highlight the status of our rebuttal **prior to the massive data-leak incident** (around 27 Nov 02:00, AoE):
- The paper received ratings of ```6,6,6,6,4``` (**5.6 on average**).
- Reviewers **t9Xo** and **1do2** explicitly confirmed that their concerns were **fully addressed** and supported acceptance (Reviewer t9Xo **raised** the score, and Reviewer 1do2 maintained the positive rating).
- The remaining negative reviewer **n8PZ**, who kept a rating of 4, also *recognized our core contribution of joint modeling principle*, but was still concerned about the simple technical design, which we addressed with comprehensive clarifications.

Below we give a concise summary of our rebuttal and clarifications to support the final assessment.

---
## Consensus Among Reviewers
- **Novelty & Contribution**: **All reviewers** highlight that our joint modeling framework identifies and addresses a critical "blind spot" in previous approaches. Reviewer **rMCB** specifically notes that we represent an "*important contribution to the knowledge of the field.*"
- **Solid Empirical Results**: Reviewers **1do2**, **3TSm**, and **rMCB** validate our empirical evaluation and acknowledge robust improvements of FMIP over SOTA baselines, describing the experiments as *comprehensive* (1do2) and *extensive* (rMCB).
- **Flexibility & Generalizability**: Reviewer **3TSm** specifically recognizes FMIP's flexibility, evidenced by its seamless compatibility with multiple GNN backbones and various downstream solvers.
- **Excellent Presentation**: Reviewers **1do2** and **rMCB** both rate the presentation as "Excellent", praising our paper's clear logical flow to explain complex concepts in a detailed yet concise manner.
---
## Summary of Revisions and Clarifications
- **Contribution Clarification (1do2, n8PZ)**: We clarify that our primary contribution is the novel **design principle of joint modeling** on continuous and integer variables, rather than merely the mixed flow architecture itself. It enables the shift to generative distribution modeling to address MILP's "one-to-many" landscape and uniquely unlocks **Holistic Guidance** (feasibility & objective), which is unattainable under previous partial modeling schemes.
- **Why Generative Models & Flow Matching (t9Xo)**: Generative distribution modeling better suits MILP's "one-to-many" solution landscape (multiple optima) than discriminative prediction. Flow matching is adopted for stability, and our contributions (Joint Modeling & Holistic Guidance) are **backbone-agnostic** and adaptable to other generative schemes like diffusion models.
- **Relationship with Exact Solvers like Gurobi (rMCB)**: We clarify that FMIP is designed to **accelerate** exact solvers like Gurobi, **not replace** them. It acts as a primal heuristic to provide high-quality warm starts.
- **Necessity of Explicit Continuous Prediction (t9Xo)**: We clarify that explicitly predicting continuous variables is practical and *computationally far superior* to recovering them via LP solving (fast matrix multiplication vs. slow iterative optimization).
- **Inference Efficiency (3TSm, 1do2, t9Xo)**: We discuss the data in Table 3, showing that FMIP's inference overhead (0.3s–4.8s) is **negligible** compared to downstream solver runtime (hundreds/thousands of seconds).
- **"w/o Continuous" Ablation (1do2, n8PZ)**: We clarify that this variant strictly disables the guidance mechanism, since $f(d,c)$ cannot be computed. It provides *direct evidence* that joint modeling is critical for FMIP's superior performance as a foundation of holistic guidance.
- **Other Revisions**: We correct some typos, clarify the training details & evaluation protocols, and remove the term "tripartite" to avoid ambiguity.
---
### Additional Experiments
- **Performance on Extremely Hard Instances (rMCB)**: We test FMIP on hard instances with up to **1,000,000 variables** on SC dataset, largely exceeding training scales. While raw Gurobi stagnates, FMIP guides the solver with high-quality warm starts, improving objective values by **orders of magnitude** or finding comparable solutions **hours faster**.
- **Adaptive Guidance Strategy (n8PZ, t9Xo)**: We design an adaptive balancing strategy on $\gamma$ to show that dynamically reweighting objective vs. constraint terms can **stabilize generation over longer trajectories**, validating a path for even better performance.
- **Scalability Breakdown (3TSm)**: We partition the MIPLIB benchmark by instance size to analyze the scalability of FMIP over different problem scales. The analysis confirms FMIP is robust on small or medium-large instances, but identifies performance degradation on ultra-large instances, showing a clear direction for future backbone scaling.

We highlight all modifications in the revised paper in blue. We believe our responses effectively address specific concerns and solidify the positive consensus.

---

### Meta-Review · Area_Chair_mcVs · 2026-01-07

**Summary:**

The paper proposes FMIP, a joint continuous–integer flow-matching framework for generating high-quality solutions to mixed-integer linear programs (MILPs) by coupling discrete and continuous variables with a joint flow matching objective, enhanced by a proposed test-time guidance for feasibility and objective improvement. Reviewers generally agree that the joint modeling and guidance design are practically valuable, and the paper is clearly written and easy to follow. The authors' rebuttal adequately addressed most concerns—especially those regarding the motivation for flow matching, bounded-integer assumptions, inference-time cost, adaptive guidance balancing, and scalability to larger datasets. The discussion trend moved toward acceptance, with at least one reviewer explicitly raising their score. Therefore, I recommend this paper for acceptance at ICLR 2026.

**Reviewer Concerns:**

Overall, the rebuttal and updated manuscript have addressed the majority of reviewers’ concerns, including the motivation for flow matching, the bounded-integer assumption, inference-time cost, and scalability to larger datasets. The discussion trend moved toward acceptance, with one reviewer (t9Xo) explicitly raising their score from 4 to 6. The main remaining concern is Reviewer n8PZ’s skepticism about novelty: they still view the flow-matching formulation as a fairly straightforward combination of existing components and therefore keep their score at 4.

**Reviewer Scores:**

I think the final scores of the reviewers are 6, 6, 4, 6, 6.

---

### Decision · Program_Chairs · 2026-01-26

Accept (Poster)